# Agrin-Matrix Metalloproteinase-12 axis confers a mechanically competent microenvironment in skin wound healing

Sayan Chakraborty [1✉], Divyaleka Sampath[1], Melissa Ong Yu Lin[1], Matthew Bilton[2,3], Cheng-Kuang Huang[2,3], Mui Hoon Nai[2,3], Kizito Njah[1], Pierre-Alexis Goy [1], Cheng-Chun Wang[1], Ernesto Guccione[1], Chwee-Teck Lim [2,3,4] & Wanjin Hong [1✉]

An orchestrated wound healing program drives skin repair via collective epidermal cell proliferation and migration. However, the molecular determinants of the tissue micro-environment supporting wound healing remain poorly understood. Herein we discover that proteoglycan Agrin is enriched within the early wound-microenvironment and is indispensable for efficient healing. Agrin enhances the mechanoperception of keratinocytes by augmenting their stiffness, traction stress and fluidic velocity fields in retaliation to bulk substrate rigidity. Importantly, Agrin overhauls cytoskeletal architecture via enhancing actomyosin cables upon sensing geometric stress and force following an injury. Moreover, we identify Matrix Metalloproteinase-12 (MMP12) as a downstream effector of Agrin's mechanoperception. We also reveal a promising potential of a recombinant Agrin fragment as a bio-additive material that assimilates optimal mechanobiological and pro-angiogenic parameters by engaging MMP12 in accelerated wound healing. Together, we propose that Agrin-MMP12 pathway integrates a broad range of mechanical stimuli to coordinate a competent skin wound healing niche.

[1] Institute of Molecular and Cell Biology, Agency for Science, Technology, and Research (A*STAR), 61 Biopolis Drive, Proteos, Singapore 138673, Singapore. [2] Department of Biomedical Engineering, National University of Singapore, 4 Engineering Drive 3, Singapore 117583, Singapore. [3] Mechanobiology Institute, National University of Singapore, 5A Engineering Drive 1, Singapore 117411, Singapore. [4] Institute for Health Innovation and Technology, National University of Singapore, 14 Medical Drive, 117599 Singapore, Singapore. ✉email: sayanc@imcb.a-star.edu.sg; mcbhwj@imcb.a-star.edu.sg

Wound healing represents a complicated yet highly orchestrated biological program restoring normalcy to damaged tissue architecture[1,2]. Cutaneous wound healing is sequentially characterized by a homeostasis phase where damage signals trigger clot formation to restrict blood flow, followed by an inflammatory phase that debrides wounded cells. Next, a proliferative phase governs proliferation, survival, and migration of keratinocytes over the wounded area in a process termed as re-epithelization. Timely execution of all the above culminates towards wound closure and renovation of tissue integrity[1,3,4]. As such, a prime factor for effective wound healing following an injury is the rate of deposition of new extracellular matrix (ECM) and its components that subsequently support keratinocyte proliferation, migration, and angiogenesis that favors the healing process[5].

In addition to the scaffolding functions, the ECM acts as a 'dynamic communicative layer' to its surrounding cells in the tissue that sustain tissue integrity[6]. The skin serves as an excellent mechanoreceptor organ to analyze how mechanical forces integrate within a structured tissue architecture to sustain key biological functions[7,8]. Underscoring this dynamic mechano-feedback between the skin cells and its surrounding ECM, the regulation of a plethora of soluble and matrix-bound proteins by the ECM is an emerging area[9]. During injury, a major chunk of ECM is lost rendering the underlying tissues incapable of responding to bulk extrinsic mechanical stress, a phenomenon called mechanoperception. The lack of mechanoperceiving ability upon inflicting an injury largely accounts for large-scale deficits in actomyosin tension affecting cell survival and migration[10,11]. As an appealing hypothesis favoring early phases of wound healing, the de novo expression of key ECM proteins within a wound-healing niche may recondition the underlying skin tissues by improving mechanoperception and potentially reinstating the mechanoreciprocity between the ECM and skin cells. On this premise, an overarching interest is to identify key ECM components that integrate mechanical stimuli to establish mechanoperception within wounded tissue, and adapting them to a wound-stressed environment.

To gain an insight into the identity of ECM proteins that may promote such mechano-feedback for a productive wound healing program, we focused on an ECM wound signature comprising of some common proteoglycans Agrin, Perlecan, and Glypicans 1–3 that are frequently overexpressed in wounded cells in vivo[26]. Among them, Perlecan and Syndecan-1 are known to be expressed by keratinocytes, promote efficient wound healing, and sustain skin tissue integrity[12,13]. Considering Harold Dvorak's seminal statement that 'tumors are wounds that never heal', it has now become increasingly appealing that an organized wound healing program, despite being self-limiting, closely parallels a tumor microenvironment[3,14,15]. In this consideration, besides a defined role in the neuromuscular junctions (NMJs)[16,17], Agrin actuates the tumor mechanotransduction network via maintaining integrin-focal adhesion integrity, activating YAP/TAZ mechanosensors, and promoting angiogenesis[18–22]. Moreover, a cardio-protective role for Agrin following myocardial infarction has also been reported[23]. Because of these exemplary roles of Agrin in the tumor and regenerative microenvironments that approximates a wound healing niche, here we show that Agrin tunes a mechanically competent wound microenvironment enforcing skin tissue healing by improving keratinocyte mechanoperception.

## Results

### Skin wounding actuates an Agrin-enriched microenvironment.
Epithelial wound healing is dramatically influenced by the re-establishment of lost ECM components to generate a new stroma that supports re-epithelization of keratinocytes facilitating wound closure[24,25]. This work was initiated by the serendipitous discovery that the expression of several ECM proteins is altered in the lineage traced wounded mouse skin epidermal keratinocytes upon mechanical injury[26]. Focusing on the early phase of the wound healing where re-epithelization is marked by Keratin 17 (KRT17) expressing keratinocytes, we detected a wound signature comprising of several ECM proteins including Agrin (AGRN), Perlecan (HSPG2), Glypican 1–3 (GPC1-3), that were enhanced in the wounded cells when compared to their non-wounded counterparts within the observed 1–10 days post-punch wound injury (Fig. 1a). Following this trend, the expression levels of these ECM proteins were validated in vitro using a scratch assay in several skin cell types including the immortalized and primary human keratinocytes (HaCaT and HEK) and human skin fibroblasts (BJ), respectively. As shown in Fig. 1b, we analyzed the mRNA and protein levels in the migratory cells that initiate wound coverage within an early time-frame lasting for 24 h. Proteoglycans AGRN, GPC1, and HSPG2 were chosen as they were significantly upregulated during the early phase of wound healing in mouse models[26] (Fig. 1a). An increase in the mRNA levels of the selected ECM wound signature gene(s) was observed across the panel of skin cells that initiated migration following mechanical wounding; out of which the expression of AGRN consistently increased in all the analyzed skin cell lines during the observed 24 h post-wounding (Fig. 1b). Of note, the Agrin expression is significantly increased in the keratinocytes HaCaT and HEK within 3–24 h post-injury, whereas significant enhancement in fibroblasts (BJ) was observed by 24 h post-wounding (Fig. 1b). In contrast, GPC1 mRNA was increased by 2–5 folds only in HaCaT cells within 3–24 h post-injury without showing a significant increment in HEK and BJ cells (Fig. 1b). HSPG2 did not show any significant increase in any of the analyzed cell lines (Fig. 1b). Consistent with the data analyzed in Fig. 1a, we observed that depleting GPC1 displayed greater inhibition of HaCaT cell migration velocities post-wounding when compared to that of Agrin knockdown (Fig. S1a–c). Despite the fact that Agrin expression was induced robustly in both keratinocytes and fibroblasts upon injury (Fig. 1b), GPC1 may also serve as an important ECM proteoglycan promoting skin wound healing. These are further substantiated by defined prior roles of Glypicans and Perlecans in skin wound healing[12,27–29]. In this study, we focused on Agrin as its role has never been documented in skin injury-related models. Besides, the protein levels of Agrin also showed a consistent ~2–4 folds increase within the observed 24 h post-wounding across the panel of human and mouse keratinocytes and fibroblasts (Fig. 1c). Even in non-wounded states, both human and mouse keratinocytes expressed higher Agrin levels when compared to their respective dermal fibroblasts, implying that keratinocytes sourced Agrin may play more critical roles in skin wound repair (Fig. 1d). In a 4 mm punch-biopsy wound healing mouse model, robust Agrin expression was detected around wound edges and in wound-beds as early as day 2, which peaked at day 4 and subsequently normalized within day 8 post-wounding (Fig. 1e). The immunohistochemical analysis further revealed a significant increase of Agrin expression within the keratinocyte layers of the epidermis in comparison to the dermis at day 2 that maximized by day 4 post-wound injury (Fig. 1f). While increased Agrin expression was also observed within the injured dermis layers between days 2 and 4, no significant change was detected in the hypodermis and dermal-white adipose tissues (D-WAT) at any stage post-injury (Fig. 1f). Together, these results revealed that Agrin expression is significantly triggered within the epidermal and dermal layers of skin upon mechanical injury.

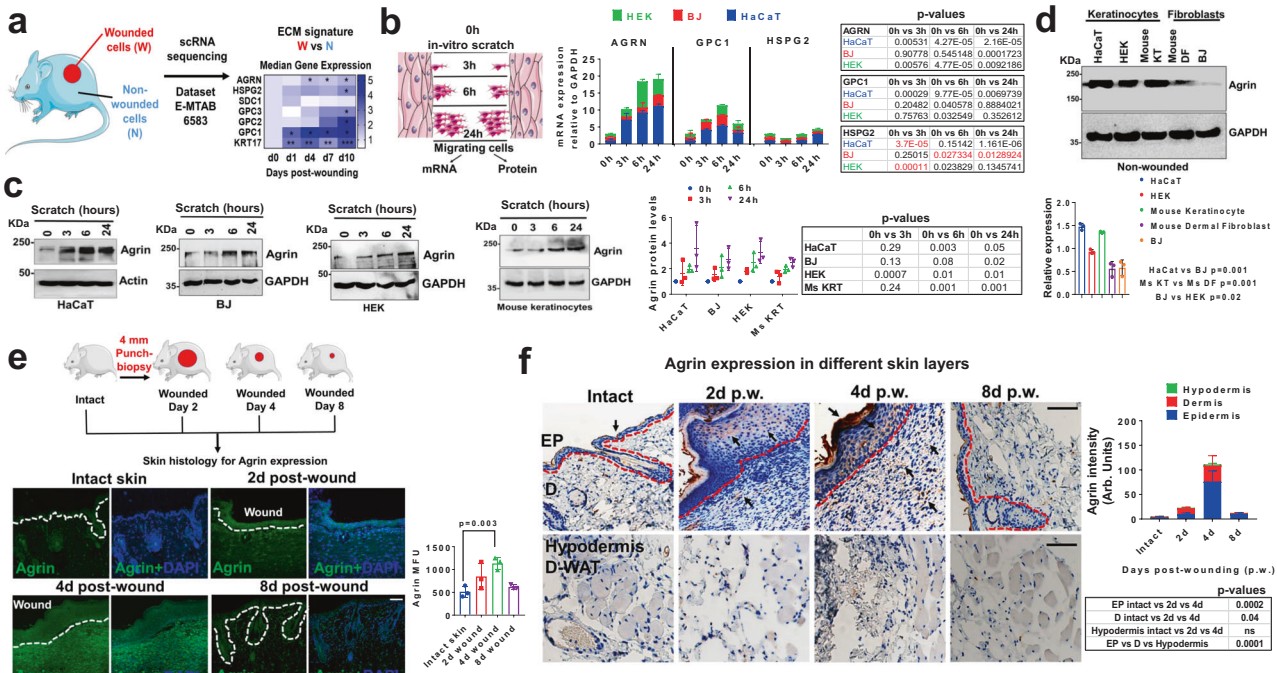

**Fig. 1 Agrin expression is enhanced during mechanical injury to the skin. a** Work-flow of gene-expression analysis of an ECM-based wound signature from single-cell transcriptomics of wounded versus non-wounded mouse keratinocytes (two-tailed Students 't' test, AGRN *$p$ = 0.05, 0.01; HSPG2, $p$ = 0.15; SDC1 **$p$ = 0.0003; GPC3 $p$ = 0.4; GPC2 $p$ = 0.15; GPC1 *$p$ = 0.01, 0.02; KRT **$p$ = 0.004, 0.002 and ***$p$ = 0.0002). **b** Confluent cells were scratched and allowed to migrate. After indicated time-points, mRNA isolated from migrating cells were analyzed by RT-PCR for the indicated genes. Data represented as mean +/− s.d., two-stage Benjamini procedure Multiple $t$-tests; $p$ values in tabular form (red values signify reduction). **c** Western blot analysis of migrating skin cells (as in panel **b**) for Agrin expression. GAPDH was used as loading controls. Densitometric quantification of Agrin levels in the indicated cells is shown graphically. Data represented as mean +/− s.d. ($n$ = 3 experiments, two-stage Benjamini Multiple $t$-tests, $p$ values shown in the figure). **d** Western blot analysis in non-wounded skin keratinocytes and fibroblasts for the indicated proteins. GAPDH was used as loading controls. Densitometric quantification of Agrin bands are presented as mean +/− s.d. ($n$ = 3 experiments, **$p$ = 0.001, *$p$ = 0.02, two-tailed Students 't' test). **e** Schematic showing the testing of Agrin expression in mouse skin post-wounding. Confocal microscopy sections of mouse skin punch-wound biopsies at indicated days were analyzed for Agrin immunofluorescence and counterstained with DAPI. Scale: 50 μm. Mean fluorescence intensity (MFU) of Agrin staining in the mouse skin sections is represented as mean +/− s.d. ($n$ = 3 mice, three sections, **$p$ = 0.003, two-tailed Students 't' test). The white dashed line denotes the epidermis and dermis boundaries. **f** Agrin immunohistochemistry of mouse skin punch-wound biopsies at indicated day post-injury. Scale: 100 μm, EP-Epidermis, D-Dermis, d-WAT-Dermal White adipose tissue. The red dotted line denotes the epidermal-dermal boundary. Agrin intensity is represented as mean +/− s.d. ($n$ = 3 mice, three sections analyzed for each time-point, two-tailed paired $t$-tests, $p$ values in the figure).

**Depletion of Agrin impairs skin wound healing**. To test the functional relevance of an Agrin-enriched microenvironment in wound healing in vivo, we utilized three independent stealth siRNAs to knockdown Agrin in the mouse skin to see the impact on healing rates following punch-biopsy wounds under 'non-splinted' and 'splinted' conditions, respectively (Fig. S2a). Stealth siRNAs offer enhanced stability, minimal off-target effects, and accessibility to skin tissues, hence are increasingly used for efficient knockdowns in animal models[30,31]. In both models, we locally injected the siRNAs at the prospective wound site three days before wounding, which efficiently reduced the basal Agrin levels at the skin injury site on the day of wounding (Fig. S2b, left panel). The scrambled or Agrin siRNAs were mixed in a topical ointment preparation and applied at the open wound site every two days to robustly deplete Agrin expression throughout the early phase(s) of healing comprising of 9 days post-wounding in the non-splinted models. At day 4, strong Agrin expression detected around wound edges and wound bed of control skin was strikingly diminished in mice ectopically treated with Agrin siRNAs which validated the efficacy of Agrin knockdown (Fig. S2c). The suppression of Agrin expression significantly delayed the in vivo cutaneous wound healing, indicating that Agrin is important for skin wound repair (Fig. 2a). Moreover, efficient wound healing is dependent on the ability of

keratinocytes to express several wound responsive Keratins such as Keratin 6, 14, and 17, that propel the formers' migratory behavior over the wound site[32]. Wound closure is initiated by a stretch of migrating keratinocytes referred to as 'epithelial tongue'. Prominent epithelial tongue predominantly concealed the wound site in scrambled siRNA treated skins while leaving the vast majority of wounds uncovered in Agrin depleted skin sections (Fig. S2d). In this vein, a significant loss (>50%) of keratinocyte migration (as shown by the relative K17 occupancy) over the wounded site was observed in mouse skin treated with Agrin siRNAs (Fig. 2b).

Furthermore, the 4 mm punch-wounds were surrounded by a 10 mm splint tightly adhered to the skin, thereby representing a 'closed' splinted mode for wound healing. The ointments containing the respective siRNAs were applied and the wound region was subsequently covered by Tegaderm (Fig. 2c). As such, the usage of splints minimized the 'purse-string' mediated wound contraction and majorly facilitated wound closure by re-epithelization[33]. Covered splinted wound dressings reduced wound healing rates at day 7 when compared to those in non-splinted conditions (Fig. 2c). Using this in vivo model as an index for measuring keratinocytes' migration, we efficiently suppressed cutaneous Agrin levels (Fig. S2b, right panels). Agrin depletion in splinted mouse models delayed skin wound healing by

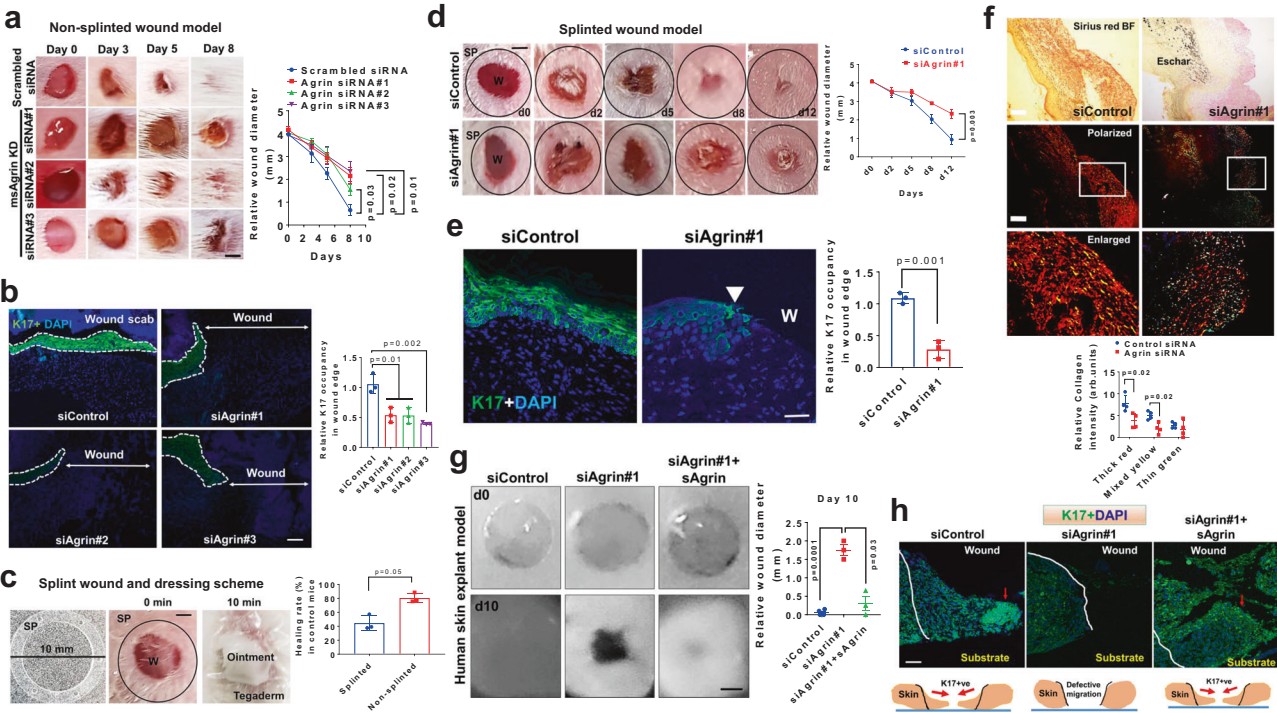

**Fig. 2 Agrin depletion impairs skin wound healing. a** Representative photographs of mouse wounds at the indicated days of treatment with siRNA containing ointments. Wound diameter presented as mean +/− s.e.m. ($n = 4$ mice per group, Two-way ANOVA, *$p = 0.02$, 0.03 and 0.01, respectively) on indicated days post-wounding. Scale: 1 mm. **b** Confocal microscopy images showing K17 and DAPI staining at day 5 post-wounding in the mouse skin sections treated as in (**a**). Scale: 50 μm. Relative K17 staining intensity represented as mean +/− s.d. ($n = 3$ sections per group, Students 't' test, two-tailed, $p = 0.01$, $p = 0.002$, respectively). **c** Photographs showing splinted wound dressing scheme. Wound diameter at day 7 in splinted and non-splinted conditions were quantified as mean +/− s.d. ($n = 3$ mice in each group, *$p = 0.05$, two-tailed Students 't' test). **d** Representative photographs of mouse splinted wounds are shown at indicated days of siRNA ointment applications. Wound diameter represented as mean +/− s.e.m. ($n = 3$ mice per group, one-way ANOVA $p = 0.003$). SP denotes splint. Scale: 1 mm. **e** Confocal images showing K17 and DAPI staining at day 7 post-wounding in the mouse skin treated with the indicated siRNAs. Scale: 50 μm. The white block arrow indicates the attenuated epithelial tongue. Relative K17 occupancy represented as mean +/− s.d. ($n = 3$ sections per group, two-tailed Students 't' test, $p = 0.001$). **f** Collagen distribution in the wound beds of control and Agrin siRNA treated mouse skin at day 7. Bright-field and polarized light images are shown. The boxed area represents the enlarged view. Scale: 100 μm. The relative collagen fibers were presented as mean +/− s.d. ($n = 4$ sections analyzed from 3 mice per group, two-tailed Students 't' test, *$p = 0.02$). **g** Representative photographs of human skin explants at days 0 and 10 post-wounding. Wound diameters at day 10 are presented as mean +/− s.d. ($n = 3$ per group, two biological repeats, two-tailed Students 't' test, ***$p = 0.0001$, *$p = 0.03$, respectively). Scale: 0.5 mm. **h** Confocal images showing K17 staining in the skin explants of control, Agrin depleted, and those rescued by sAgrin ($n = 3$ sections per group from three explants, repeated twice). The red arrow marks the leader keratinocytes. The white line indicates the wound boundary. Scale: 50 μm.

attenuating keratinocyte re-epithelization as shown by reduced K17 occupancy (Fig. 2d, e). Coupled to impaired re-epithelization, Agrin depletion severely dampened the deposition of mature and intermediate collagen fibers in the wound beds, indicating that compromised ECM replenishment in Agrin depleted skin is associated with delayed healing response (Fig. 2f).

Moreover, we extended our study using siRNAs against human Agrin to inhibit its expression in keratinocyte and in a human skin explant model (Fig. S2e, f). Notably, Agrin expressing keratinocytes effectively migrated to close the wound in control siRNA treated skin explants, which was inhibited by Agrin siRNA#1 (Fig S2f, g). The depletion of Agrin (by Agrin siRNA#1) robustly attenuated the K17 expressing epithelial tongue migration and thereby clearly retarded the ex vivo wound closure rates in these human skin explants (Figs. 2g, h, S2g). Interestingly, supplementing the recombinant C-terminal fragment of Agrin (sAgrin) harboring the binding sites to its receptors Lipoprotein related receptor-4 (Lrp4) and integrins significantly rescued the wound healing and the migration of keratinocytes in the human skin explant model (Fig. 2g, h). In accordance to these in vivo and ex vivo functions, knockdown of Agrin severely impaired the trans-well migration and scratch wound-responsive migration

velocities in dermal fibroblasts and keratinocytes (of human and mouse origin) that was again in part, rescued by sAgrin (Fig. S3a–d).

As keratinocyte re-epithelization is tightly coordinated with their proliferative states[34], we next investigated whether Agrin supported keratinocytes' proliferation during their migratory phase post-wound injury. During early in vitro migration within 4 h post-wound injury, Agrin depletion did not affect the proliferative rates of the leader cells as measured by 5-bromo-2'-deoxyuridine (BrdU) incorporation assays (Fig. S4a). However, the proliferation of follower cells away from the wound margin was significantly reduced by Agrin knockdown, an effect which was rescued by sAgrin supplementation (Fig. S4a). Thus, Agrin depletion differentially affects the proliferative states of leader and follower cells in vitro. In vivo, an efficient wound re-epithelization is achieved when highly proliferative keratinocytes (followers) at the wound edge push the migrating cells (leaders) over the wound[35] (Fig. S4b). As shown in the control siRNA-treated wound sections, proliferative keratinocytes at the base of the epidermis (yellow arrows) are actively pushed during re-epithelization by the hyper-proliferative wound edge regions (red arrows) at day 7 (Fig. S4b). Coupled to impaired

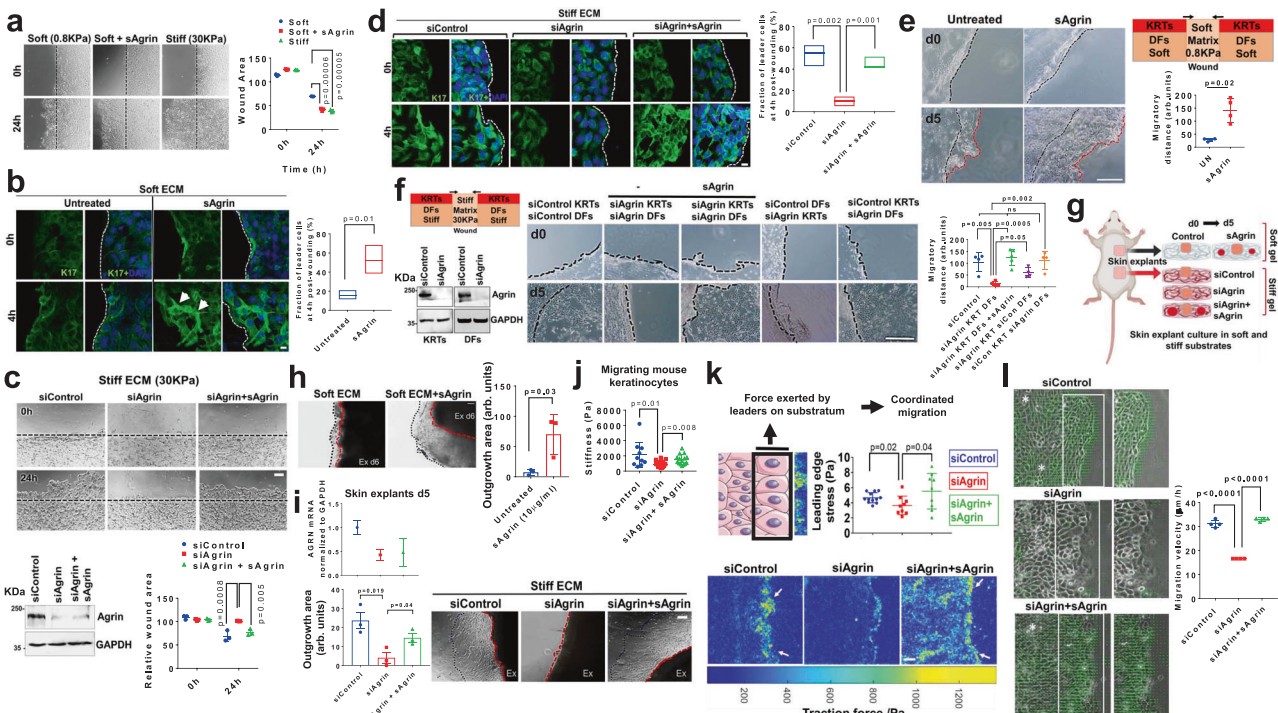

**Fig. 3 Agrin sensitizes rigidity-dependent keratinocytes migration by enhancing traction and velocity. a** Bright-field images of HaCaTs on soft or stiff substrates with/without sAgrin for 18 h. Wound area presented as mean +/− s.d. ($n = 3$, Benjamini Multiple-t test, $p = 0.00006, 0.00005$). Scale: 50 μm. **b** Confocal images showing K17 and DAPI in HaCaTs as in (**a**). Scale: 10 μm. K17 intensity was quantified ($n = 10$ per group, Students 't' test, *$p = 0.01$, bars-1-99 percentile, central lines-median). White arrows- leader cells. Dashed line-wound edge. **c** Bright-field images of control, Agrin depleted or rescued HaCaTs on 30 KPa. Migratory area quantified as mean +/− s.d. ($n = 3$, Two-way ANOVA). Scale: 50 μm. Western blot showing Agrin knockdown. **d** Confocal images of K17 and DAPI in cells treated same as in (**c**). Scale: 10 μm. Dashed line shows wound edge. K17 intensity is quantified ($n = 10$ per condition, Students 't' test, **$p = 0.002, p = 0.001$, Box-1-99 percentile, lines-median). **e** Schematic and bright-field images of 3D-migration with/without sAgrin on 0.8 KPa. Scale: 100 μm. ($n = 4$ per group, Students 't' test, *$p = 0.02$). Black lines-original wound; red-migrated region. **f** Western blot showing Agrin knockdown in cells used for 3D-migration. Bright-field images of migratory cells with/without sAgrin on stiff matrix. Scale: 100 μm. Migration area presented as mean +/− s.d. ($n = 4$ per group, Students 't' test, $p = 0.05, 0.005, 0.002, 0.0005$). **g** Mouse skin explant set-up. **h, i** Bright-field images of keratinocytes on 0.8 KPa with/without 20 μg/ml sAgrin (**h**). Skin explants in control, Agrin depleted or conditioned with sAgrin on 30KPa (**i**). RT-PCR for Agrin (**i**) is shown as mean +/− s.d. ($n = 3$ repeats). Outgrowth area represented as mean +/− s.d., $n = 3$ per group, Students 't' test, *$p = 0.03$ (**h**), *$p = 0.019$ and $0.04$ (**i**). Scale: 50 μm. AFM (**j**) and TFM (**k**) in Control HaCaTs, Agrin depleted and those rescued with sAgrin at 4 h post-wounding. **j** Mean stiffness (Pascals) +/− s.d. are shown ($n = 10$ cells, three replicates, Students 't' test, *$p = 0.01$, *$p = 0.008$). White arrows-leading edge. Scale: 100 μm. **k** Traction forces shown as mean +/− s.d. ($n = 6$ images, three experiments, Students't' test, *$p = 0.02$ and $0.04$). **l** PIV in HaCaTs treated as in (**j**). Green arrows-velocity. White asterisks-vortex. Scale: 100 μm. Velocities presented as mean +/− s.d. ($n = 3$, Students 't' test, ***$p = 0.00001$). Statistics are all two-tailed.

proliferation at the base of epidermis, Agrin depletion severely hampered the keratinocyte proliferation in the dermal layers at the wound edge and beds (Fig. S4b, c). Therefore, these shreds of evidence suggest that Agrin enriched in the wound environment supports keratinocyte proliferation, which primes an efficient collective migration phase and wound closure.

**Agrin sensitizes keratinocytes towards ECM rigidity and fluidic collective migration.** Having demonstrated a role in wound healing response, we explored whether Agrin generates a mechanically competent environment favoring collective keratinocyte migration and wound closure. Since bulk stiffness from the ECM stimulates the migration of a variety of cell types[10], we rationalized that Agrin may integrate ECM stiffness signals with collective keratinocyte migration. Collective migration of HaCaT cells cultured on stiff (30 kPa) substrates was significantly higher than those on compliant ones (0.8 kPa) (Fig. 3a). Interestingly, incorporation of sAgrin within compliant substrates significantly promoted migration rates to levels comparable to stiff ECM (Fig. 3a). The collective keratinocyte migration is initiated by the expression of Keratin 17 (K17), particularly by the leader cells located at the wound edges[35,36]. In such conditions, supplementing sAgrin to soft substrates stimulated collective migration by activating K17 expressing leader cells (Fig. 3b). In contrast, compared to control cells, Agrin depleted keratinocytes on stiff substrates had slower migratory potential and this was significantly rescued when sAgrin was integrated into the stiff matrix (Fig. 3c). Agrin depletion in cells experiencing stiff substrates also resulted in a dramatic loss of migrating leader cells expressing K17 (Fig. 3d). The fraction of K17 expressing leader cells was significantly restored in Agrin depleted cells upon sAgrin supplementation in the stiff ECM that correlated with enhanced migration (Fig. 3c, d).

Because 2D-scratch assays using single-cell types do not reflect the physiology of the 3D wound environment experienced during injury, we next devised an in vitro 3D-substrate stiffness dependent migration assay to recapitulate the role of Agrin in supporting keratinocytes' migration under the simultaneous influence of underlying dermal cultures and bulk substrate rigidity as experienced by native skin tissues. We first embedded primary mouse dermal fibroblasts (DFs) within a compliant

collagen matrix (0.8KPa) which were subsequently overlaid with mouse keratinocytes (KRTs) (Fig. 3e). A wound was created at the center and the resulting area was replenished by a soft matrix immediately allowing the keratinocytes to migrate through a compliant environment. As shown in Fig. 3e, supplementing sAgrin within the soft matrix strongly enhanced keratinocyte migration by day 5 post-injury. Consistently, Agrin depletion in both KRTs and DFs suppressed the migratory rates when experiencing stiff matrix conditions (Fig. 3f, second panel). Interestingly, supplementing sAgrin in this condition strongly restored keratinocyte migration for the cells depleted of Agrin (Fig. 3f, third panel). Of note, Agrin expressed by the DFs was not sufficient to rescue the migration of Agrin depleted keratinocytes over the stiff substrates (Fig. 3f, fourth panel). Furthermore, Agrin depletion in DFs had minimal effects on keratinocyte migration over stiff substrates (Fig. 3f, fifth panel). Together, these data indicate that increased Agrin expression within keratinocytes majorly responds to ECM rigidity sensing, which is necessary and sufficient to drive migration in the wound microenvironment.

We next tested the ability of sAgrin to sensitize collective ex vivo keratinocyte outgrowth from mouse skin explants experiencing varied substrate rigidities. Accordingly, collective keratinocyte outgrowth was monitored when full-thickness hind skin from a 2-day old mouse pup was placed on either collagen matrix corresponding to soft or stiff substrates in the presence or absence of sAgrin, respectively (Fig. 3g). Compared to the explants on soft collagen matrices alone, sAgrin supplemented compliant substrates significantly enhanced keratinocyte outgrowth (Fig. 3h). Agrin depletion by mouse-specific siRNA treatment in these skin explants experiencing a stiff substrate without any exogenous sAgrin failed to generate keratinocyte outgrowth after five days post-culture (Fig. 3i, middle panel). Importantly, Agrin depleted skin explants that were sensitized by sAgrin within stiff substrates exhibited higher keratinocyte outgrowth (Fig. 3i, third panel). Cumulatively, these data suggest that Agrin empowers ECM rigidity sensing in keratinocytes that guides collective migration post-wounding.

Cells respond to bulk ECM stiffness by introducing subtle changes to their intrinsic mechanical properties by enhancing their cytoskeletal tension favoring greater motility[10,37]. This mechanoperception has been documented in tumor-initiating cells[38], however, the response initiated within keratinocytes under wound stress remains elusive. To examine the mechanoperception of keratinocytes bestowed by an Agrin-enriched environment, we first measured the stiffness of migrating keratinocytes post-injury by Atomic Force Microscopy (AFM) as a marker for cell-intrinsic mechanical property. The stiffness of migrating mouse keratinocytes recorded as ~2.1 KPa at 4 h post-scratch was significantly decreased to ~0.9 KPa upon Agrin knockdown (Figs. 3j and S5a, b). Notably, sAgrin attributed a significant increase in cell stiffness (~1.5 KPa) for knockdown cells, thereby enhancing mechanoperception of these migrating keratinocytes (Figs. 3j and S5b). As such, Agrin knockdown 'softens' the migratory cell making them incompetent to navigate across wound bed. Next, we performed traction force microscopy (TFM) that measures the cumulative forces exerted by migrating cells on the substrate in a mechanically stressed environment mimicking a wound injury[37,39]. In this collective migration model, the maximal traction force is exerted by the cells at the leading edge on the ECM[40,41]. The collective migration of Agrin depleted cells were severely hampered illustrating lower mean traction stress when compared to the control cells (Fig. 3k). While the control cells had higher collective traction forces (yellow clusters) at the leading edges, Agrin depleted ones had significantly lesser traction stress within the leader and follower cells resulting in lower migratory potential (Fig. 3k and Supplementary Videos 1,

2). The average traction stress of leader cells after 6 h post-migration was ~4.67 Pa, which was reduced by ~32% upon Agrin depletion. Also, we mapped the velocity field using particle image velocimetry (PIV) to characterize the fluidic dynamics of collective cell migration in epithelial wounds[41–43]. Homogenous velocity distribution was observed in control cells attributed by large magnitude velocity vectors with occasional swirling motions and vortices presenting a fluid-elastic-like migratory behavior (Fig. 3l and Supplementary Video 1). This fluidic migration was largely disrupted by the suppression of Agrin (Fig. 3l and Supplementary Video 2). Consistently, collective traction stress at the leading front, coordinated velocity fields, and fluidic collective migration were significantly enhanced by supplementing sAgrin to Agrin depleted cells (Fig. 3k–l and Supplementary Video 3). These results illustrate that Agrin confers cell-intrinsic stiffness and enhances traction to ECM attributing coordinated fluidic dynamics to collective keratinocyte migration.

**Agrin mechanotransduction tunes cell mechanics post-injury.** Central to the outcome of enhanced keratinocytes' fluidic motility, we asked whether Agrin tunes cellular mechanics during wound injury via coordinating cytoskeletal architecture. An organized cytoskeletal architecture determining the integrity of collective migration in keratinocytes is showcased by the formation of actomyosin cables at the leading front in embryonic and adult wound healing[41,44–46]. First, we examined whether Agrin orchestrated actomyosin dynamics at the leading edge during wound stress. Wounding generated robust actomyosin cables within 4 h in control keratinocytes (Fig. 4a). As expected, Agrin deprived cells lacked these actomyosin cables, which were restored by exogenously added sAgrin (Fig. 4a). This actomyosin cable network is underlined by the dramatic induction of phosphorylated myosin light chain (pMLC) within 2 h post-wound injury in control cells that bestows enhanced contractility and migration velocity[46] (Fig. 4a, b). Agrin depleted cells consistently showed a severe dampening of pMLC activation within 2 h post-wounding, and this was restored by pre-treatment with sAgrin (Fig. 4b). Recombinant sAgrin treatment restored strong recruitment of pMLC to wound edges forming stable actomyosin cables that were abolished by the MLC inhibitor, blebbistatin (Fig. 4c). Second, the formation of robust actomyosin cables was integral to collective cell migration as blebbistatin treatment abolished the sAgrin induced migration of HaCaTs post-wound injury (Fig. 4d). These lines of evidence support that Agrin organizes actomyosin architecture at the leading front for sustaining the migratory potential of cells post-wounding.

To gain insights into how cellular mechanoperception is calibrated through actomyosin engagement by an Agrin-induced force recognition mechanism, we used a permanent magnet to apply mechanical force on wounded keratinocytes via ligand coated magnetic beads that simulated the effects of abnormal mechanical force experienced by the wound microenvironment. In this setup, magnetic beads were conjugated with control proteins (Bovine Serum Albumin-BSA or Fibronectin-FN) and sAgrin, respectively, and were subsequently allowed to bind to the cell surface. A permanent magnet placed 6 mm above the cells in culture plate ensured a sustained force of 200 pN was applied for 30 min post-wound scratch through the ligand coated beads (Fig. S6a)[47,48]. The effective ligand conjugation to the beads via covalent bonding is shown by the abrupt reduction of remnant free proteins post-conjugation (Fig. S6b). Similar to the integrin ligand FN, sAgrin beads substantially increased activated integrin β1-Agrin localization in control migrating keratinocytes, suggesting an enhanced mechanotension is induced upon localized force application (Fig. S6c). We detected actomyosin cables at 30 min

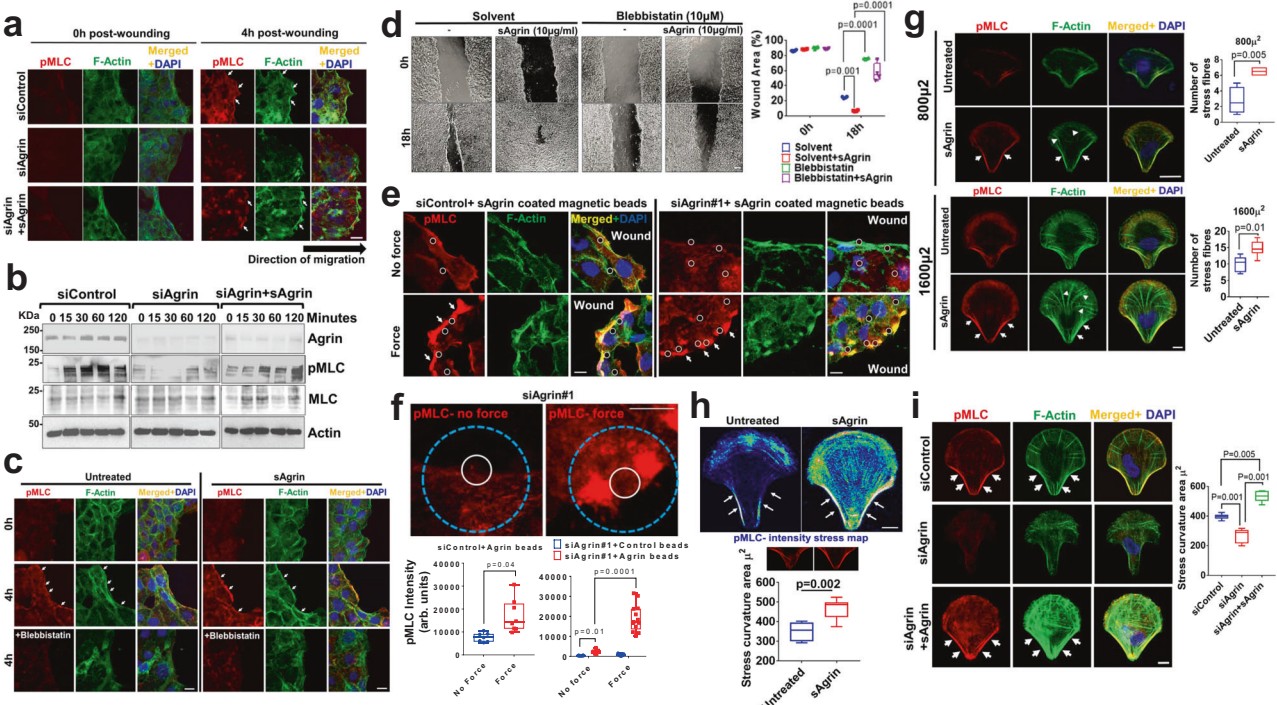

**Fig. 4 Agrin tunes cell mechanics during wound injury. a** Confocal images showing pMLC and F-Actin staining in control HEK, Agrin depleted and rescued with 10 µg/ml sAgrin at the indicated time post-scratch. Scale: 10 µm. White arrows show acto-myosin cables. **b** Western blot analysis in HEK treated as in (**a**) post-scratch injury after the indicated time (n = 3 biological repeats). **c**, **d** Confocal images of pMLC, F-Actin, and DAPI in untreated or sAgrin nourished HEK in the presence or absence of blebbistatin treatment (2 h). Scale: 10 µm (**c**). Bright-field images of migrating cells at 18 h post-scratch (**d**). Wound area is plotted in box-whiskers (n = 3, two-tailed Students 't' test, **p = 0.001, ***p = 0.0001 respectively, Box-1-99 percentile, whiskers-minimum to maximum, central line-median). Scale: 100 µm. **e** Confocal images showing pMLC, F-Actin, and DAPI staining under normal or magnetic force-recognition by BSA or sAgrin beads in Control and Agrin depleted HEKs. Scale: 10 µm. White circles indicate beads. White arrows indicate pMLC-Actin cables. **f** Mean pMLC intensity around sAgrin beads in Agrin depleted HEK cells presented from three experiments (n = 8 cells, two-tailed Students 't' test, *p = 0.04, ***p = 0.0001, Box-1-99 percentile, whiskers-minimal to maximum values, central line-median). Scale: 10 µm. The white circle indicates beads. The blue circle represents 200 µ² area around each bead. **g** Representative confocal images of indicated HEKs in small or large micropatterns with/without 20 µg/ml sAgrin for 4–6 h showing pMLC and F-actin distribution with DAPI. Stress fiber was quantified as mean +/− s.d. from three experiments (n = 10 cells per group, two-tailed Students 't' test, **p = 0.005, *p = 0.01, respectively, Box-1-99 percentile, whiskers- minimum to maximum values, central line-median). Scale: 10 µm. **h** Confocal images of pMLC intensity of HEKs on large fibronectin or sAgrin-coated micropatterns. Scale: 10 µm. Stress curvature is presented from three experiments (n = 10 cells per group, two-tailed Students 't' test, **p = 0.002, Box-1-99 percentile, whiskers-minimum to maximum values, central line-median). **i** Confocal images of Control or Agrin depleted HEKs cultured same as in (**h**) were immunostained with pMLC and F-actin with DAPI. Scale: 10 µm. Stress curvature presented from three experiments (n = 10 cells per group, two-tailed Students 't' test, **p = 0.001, **p = 0.005 Box-1-99 percentile, whiskers- minimum to maximum values, central line-median).

post-scratching in control cells without any additional magnetic force (Fig. 4e, no force panel); however, application of exogenous force further enhanced pMLC recruitment localized near sAgrin beads forming robust actomyosin cables (Fig. 4e, force panel). In stark contrast, Agrin knockdown cells exhibited poor localization of pMLC and actomyosin cables under normal conditions post-wounding, as reported in Fig. 4a, respectively (Fig. 4e). We anticipated that transient force transmission by sAgrin, in part, should render greater actomyosin recruitment as a manifestation of enhanced mechanoresponse of wounded cells. Accordingly, application of force with sAgrin for 30 min led to a ~2 folds increase of pMLC around the bead vicinity that partially rescued the actomyosin cables activity in Agrin depleted cells without any prior exposure to sAgrin in the culture media (Fig. 4e, f). Upon force application to wounded cells, sAgrin induced greater pMLC recruitment in control cells compared to FN, but similar to that induced by Syndecan-4 (SDC4) (Fig. S6d, e, siControl panels). Although FN stimulated integrins as global mechanosensors[49] and syndecan-4 tuned cell mechanics[47], these were rather insufficient to rescue myosin mechanotension in Agrin deprived cells. Essentially, the reduced pMLC in Agrin depleted cells was

only restored by force transduced by sAgrin (and not by FN and SDC4) beads (Fig. S6d, e, siAgrin panel). Likewise, compared to FN or BSA beads, sAgrin coated beads enhanced pMLC recruitment upon force application in Agrin depleted cells at 30 min post-wounding (Fig. S6f–h). The specificity of Agrin as a mechanotransducer was additionally tested by the following experiments: first, increasing sAgrin coated beads exerted greater pMLC recruitment at 30 min post-wound scratch in a dose-dependent fashion (Fig. S6i). Second, temporal increase of force application via constant stimulation by sAgrin coated beads up to 60 min led to an enhanced pMLC mechanoperception following wound-injury (Fig. S6j). These results advocate that Agrin acts as a mechanotransducer of extrinsic force to stimulate cytoskeletal dynamics required for collective migration following wound injury.

In addition to the loss of ECM components, keratinocytes often navigate through wounded areas adapting to large-scale changes to their morphology and cytoskeleton under different geometrical tensions[50]. To simulate whether Agrin influences keratinocytes' ability to shift their cytoskeletal tension upwards upon exposure to geometrical constraints, we cultured normal HEK cells in

crossbow-shaped FN patterns of different surface areas. The FN coated crossbow micropatterns force the cells to assume a polarized orientation with F-actin stress fibers originating from the dorsal arc and extensive actomyosin stress fiber bundling at the transverse arc towards the base[51,52]. Smaller (800 μ²) fibronectin patterns compressed cells even further leading to severe loss of F-actin stress fibers (Fig. 4g). Despite within small confinements, sAgrin supplemented crossbow patterns partially rescued F-actin stress fibers with higher pMLC recruitment to the transverse arc (Fig. 4g). This effect of sAgrin induced stress fiber formation in geometrically constrained HEK cells was further accentuated in large crossbow (1600 μ²) patterns, where cells demonstrated a prominent tension signature comprising of elongated F-actin stress fibers and actomyosin tension at the transverse arc (Fig. 4g, lower panels). Of note, sAgrin conferred significantly higher tension curvatures that were enriched with activated MLC at the transverse arcs of cells in large crossbow patterns (Fig. 4h). Importantly, F-actin stress fibers and pMLC enriched transverse arc tension bundles were essentially abrogated in Agrin depleted cells (Fig. 4i, first and second panels, respectively). Interestingly, sAgrin incorporated within the FN

crossbow matrix restored the tension signature of F-actin stress fiber network and pMLC enriched transverse arcs in cells depleted of endogenous Agrin (Fig. 4i, third panel). Together, these results suggest that tension sensing mechanisms are conferred by Agrin in keratinocytes under geometric constraints, thereby adapting them towards an ECM matrix that is favorable for the healing program.

**MMP12 as a mediator of Agrin-mechanotransduction following wound injury.** To identify the downstream effectors of Agrin-mediated mechanotension in keratinocytes upon wound injury, we performed transcriptome-wide comparison via RNA-seq analysis in control versus Agrin depleted cells cultured in plastics that mimic a highly stiff ECM (Fig. S7a). Multi-dimensional scaling (MDS) revealed distinct gene profiles between the control and Agrin depleted bulk RNA population (Fig. S7a). Upon stratifying a staggering number of genes that were differentially regulated by Agrin in stiff ECM, a few highly significant clusters using gene ontology (GO) analysis were selected (Figs. 5a, S7b, and Supplementary Data 1, 2). As

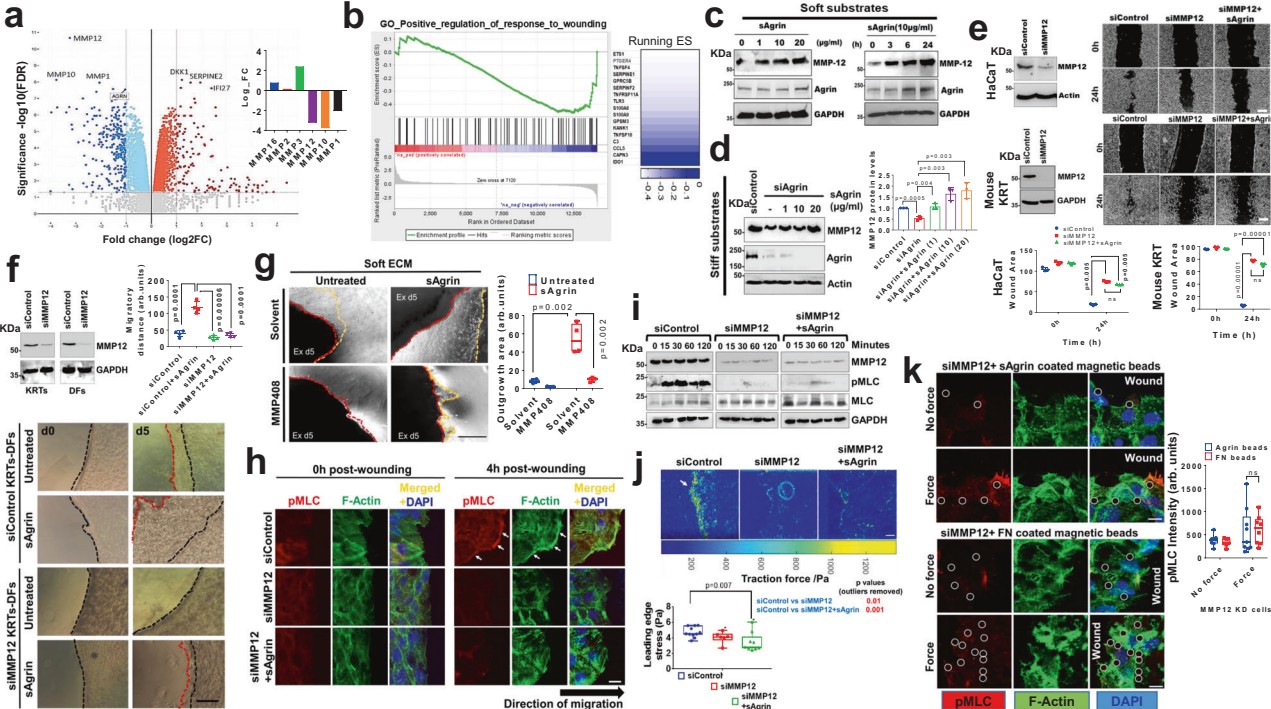

**Fig. 5 MMP12 potentiates Agrin's mechanoperception. a** Volcano plot of differentially expressed genes in si-control and si-Agrin HaCaTs. Inset showing MMP genes. (n = 3 replicates). **b** GSEA showing the downregulated gene cluster. **c** Western blot for the indicated proteins from HaCaT cells on 0.8 KPa under varied sAgrin concentrations for 18 h (left) or 10 μg/ml sAgrin for the varying time (right). **d** Western blot for the indicated proteins in control, Agrin depleted and rescued HaCaT on 30 KPa. MMP12 bands were quantified as mean +/− s.d. (n = 3, Two-stage, Holm−Sidak Multiple t-tests). **e** Western blot detecting MMP12 levels in indicated cells. Bright-field microscopy images of same cells post-scratch assay. Wound area presented as mean +/− s.d. (n = 3, two-tailed Students 't' tests, **p = 0.005, ***p = 0.0001, respectively). Scale: 50 μm. **f** Western blot showing MMP12 knockdown (n = 2). Bright-field images of control and si-MMP12 cells migrating with/without 10 μg/ml sAgrin on 0.8KPa. Scale: 100 μm. Black and red dotted lines show the initial wound and end-migrated areas. Migration area was quantified as mean +/− s.d. (n = 4 wounds, two-stage Benjamini Multiple 't' tests, ***p = 0.0001 and ****p = 0.00006). **g** Mouse skin explants on 0.8 KPa with/without 20 μg/ml sAgrin were treated with DMSO or MMP408 (5 nM). Bright-field images showing outgrowth on day 5. Scale: 50 μm. Outgrowth area was quantified (n = 4 per group, Two-tailed Students 't' test, **p = 0.002, Box-1-99 percentile, whiskers-minimum to maximal values, central line-median). **h** Confocal images showing p-MLC, F-Actin, and DAPI staining in control or MMP12 silenced HaCaTs in the presence or absence of sAgrin. Scale: 10 μm. White arrows show actomyosin cables. **i** Cell lysates from HEK treated as in (**h**) were analyzed by Western blot (n = 3 repeats). **j** traction force map in control and si-MMP12 HaCaTs with/without sAgrin (10μg/ml) for 18 h. Scale: 100 μm. Mean traction stress (in Pascals) are shown (n = 6 images from three experiments, two-tailed Students't' test, *p = 0.007, Box- 1-99 percentile values, whiskers-minimum to maximum, central line-median). **k** Confocal images showing pMLC-F-Actin and DAPI staining upon sAgrin/Fibronectin-induced force in MMP12-depleted HaCaTs. Scale: 10 μm. White circles represent beads. Mean pMLC intensity +/− s.d. was quantified from three experiments (n = 10 cells per group, two-tailed Students 't' test, ns, Box-1-99 percentile, whiskers- minimum to maximum values, central line-median).

expected, GO, network and Gene set enrichment analysis (GSEA) revealed that the majority of genes that were drastically down-regulated upon Agrin knockdown belonged to ECM modulation including matrisome-associated and structural components of ECM (Fig. S7b–d). Among the top 10 significant genes in response to Agrin knockdown, several matrisomal proteins including the matrix metalloproteases (MMPs) were consistently reduced while others such as Serpine2, Interleukin 1a-b, were upregulated upon Agrin depletion (Figs. 5a and S7e). In line with effects on cell migration, GSEA analysis also revealed a global reduction in gene cluster that positively regulated wound healing (Fig. 5b). We validated the mRNA expression(s) of the selected set of up- and downregulated genes (Fig. S7f). Expression of mRNAs of several MMPs including MMP12, 10, and 1 was found to be significantly reduced upon Agrin knockdown.

Our next set of experiments focused on identifying the key MMPs that mediate Agrin's mechanical functions in the wound environment. First, we evaluated whether MMP 1, 10, and 12 independently affected keratinocyte migration following wound-scratch. Accordingly, knocking down MMP 1 and 10 reduced keratinocytes' migration that was not restored by sAgrin treatment, suggesting they are involved in Agrin-mediated functions (Fig. S8a). Importantly, MMP12 knockdown showed the maximal inhibition of migration velocities post-injury (Fig. S8b). The mRNA levels of MMP1 and MMP10 were not consistently upregulated following wound injury in HaCaT cells (Fig. S8c). Most importantly, MMP1 or MMP10 depletion had no effects on the acto-myosin tension at the wound edges (Fig. S8d). When cultured on large crossbow patterns, MMP1 or MMP10 depleted cells did not impact the tension signatures of F-actin stress fiber network and pMLC enriched transverse arcs (Fig. S8e). Despite being regulated by Agrin, these observations suggest that MMP1 or 10 are not major downstream mediators in restoring the wound healing mechanics.

Despite the fact that several MMPs are involved in cancer cell migration, matrix crosslinking, and wound healing, a concrete role of MMP12 has not been elucidated in mechanotransduction involved during cutaneous wound repair. As such, MMP12 emerged as the most significant effector for mediating Agrin's mechanotension in wound repair. Similar to its mRNA levels, MMP12 protein levels were significantly reduced upon Agrin depletion in a panel of immortalized and primary keratinocytes and dermal fibroblasts cultures (Fig. S9a). Due to a decrease in both mRNA and protein levels upon Agrin depletion, a loss of MMP12's gelatin and casein degradation catalytic activity was obvious in Agrin silenced keratinocytes, which was restored by sAgrin supplementation in a dose-dependent pattern (Fig. S9b).

We next examined whether MMP12 activation is a consequence of Agrin's sensitization towards ECM rigidity in a wound injury-associated Agrin-rich environment. First, we cultured HaCaT cells in soft substrates (0.8 kPa) alone or with increasing concentration of supplemented sAgrin. Importantly, soft substrates incorporated with sAgrin strongly stabilized MMP12 protein levels in a dose-dependent fashion (Fig. 5c, first panel). Enhancing the exposure time of keratinocytes cultured in soft matrix conditioned with sAgrin also increased MMP12 levels, suggesting that Agrin closely mimics ECM rigidity cues to positively regulate MMP12 levels in keratinocytes (Fig. 5c, right panel). On the contrary, the depletion of Agrin in cells experiencing stiff ECM led to a significant reduction in MMP12 levels (Fig. 5d). Agrin-depleted cells restored their MMP12 levels upon sensing sAgrin cues from the stiff ECM in a dose-dependent manner (Fig. 5d). Second, unlike MMP1 or 10, wound injury strongly activated mRNA (~4 folds) and protein levels of MMP12 that closely mirrored that of Agrin increase in response to injury of several skin cells (Figs. S9c, S8c and Fig. 1b). As early as 6 h

post-scratch, MMP12 and Agrin were detected within cells at the leading migrating front; this effect was further heightened by sAgrin supplementation, hence, simulating a favorable platform for collective keratinocyte migration following wound injury (Fig. S9d). The migrating epithelial tongue mediating in vivo wound closure also expressed MMP12 in control siRNA-treated mouse skin (Fig. S9e). Localized Agrin depletion significantly deprived MMP12 levels within the attenuated epithelial tongues that subsequently failed to close the wound (Fig. S9e). In our splinted wound healing in vivo models, Agrin depletion suppressed MMP12 expression within day 4 post-injury associated with poor healing rates (Fig. S9f). This was recapitulated in vitro where MMP12 knockdown mimicked the loss in migration of human immortalized and primary mouse keratinocytes post-scratch similar to that exhibited by Agrin depleted cells (Fig. 5e). Cumulatively, these shreds of evidence suggest that a wound-stressed Agrin microenvironment activates MMP12 for collective keratinocyte migration.

Several snippets of observations further imply that MMP12 mediated the Agrin-induced collective migration post-wounding. Firstly, sAgrin failed to stimulate migration in MMP12 depleted human and mouse keratinocytes, suggesting that MMP12 is an essential effector mediating Agrin's functions (Fig. 5e). Secondly, knockdown of MMP12 in keratinocytes experiencing stiff substrates (30 kPa) severely hampered K17 expressing migrating cells, and this was not restored by the nourishment of sAgrin (Fig. S9g). Thirdly, short-term treatment of keratinocytes with MMP408, a specific MMP12 inhibitor, blunted the migration of sAgrin induced K17 expressing keratinocytes in soft substrates and under normal cell culture conditions (Fig. S9h, i). Fourthly, sAgrin incorporated soft matrix stimulated mouse keratinocyte migration over dermal fibroblasts post-injury as reported previously in 3D stiffness-dependent migration assays (Fig. 5f, first and second panels). In contrast, depleting MMP12 suppressed the role of sAgrin in compliant matrices (Fig. 5f, third and fourth panels). Ex vivo, the enhanced K17 expressing migratory outgrowth from the mouse skin explants induced by sAgrin in soft matrix was abolished by the presence of the MMP408 inhibitor (Figs. 5g and S9j). Collectively, these results suggest MMP12 is a downstream effector essential for multiple roles of Agrin.

To further establish MMP12 as a mediator of Agrin's mechanotension in wound-stressed keratinocytes, the next set of experiments was focused on determining the role of MMP12 in overhauling cytoskeletal tension in response to an Agrin-enriched early wound microenvironment. The robust actomyosin cables generated in Agrin complacent keratinocytes post-wounding were severed by the depletion of MMP12 that accompanied with pronounced inhibition of pMLC at the wound edges (Fig. 5h). Even sAgrin treatment did not restore actomyosin cable network unmasking a conspicuous defect in cytoskeletal orientation in these MMP12 suppressed cells (Fig. 5h). Likewise, sAgrin treatment was unable to rescue the severely diminished pMLC levels in MMP12 depleted keratinocytes within the observed 2 h post-wounding (Fig. 5i). Moreover, compared to the control cells, MMP12 knockdown cells and those with sAgrin nourishment did not exhibit F-actin stress fibers and pMLC enrichment in the transverse arcs when geometrically stretched in large crossbow patterns (Fig. S9k), suggesting that overhauling cytoskeletal tension by Agrin during collective cell migration post-wound injury requires functional MMP12.

Due to impaired collective migration upon MMP12 inhibition, we examined whether reduced cytoskeletal tension is a consequence of progressive weakening in the mechanotransducing abilities of Agrin arising from MMP12 inhibition. To this end, we first mapped the cell stiffness of MMP12 suppressed mouse keratinocytes during

at 4 h post-injury. Strikingly, MMP12 knockdown significantly reduced the stiffness of migrating keratinocytes which were not restored by sAgrin treatment (Fig. S9l). In addition to reduced cell stiffness, the traction stresses of MMP12 knockdown keratinocytes migrating on TFM substrates were partially reduced at the migrating front (Fig. 5j). Interestingly, the treatment of sAgrin failed to induce sufficient traction force at the migrating leading edge of MMP12 depleted keratinocytes (Fig. 5j). Compared to control cells, MMP12 knockdown inhibited the homogenous velocity distribution at the leading front, which further decreased the fluid-elastic-like migration in sAgrin supplemented cells (Fig. S9m and Supplementary Videos 4–6). These results suggest that Agrin engages MMP12 to provide a fluidic migratory behavior to keratinocytes. Furthermore, the pMLC activation via Agrin induced force transmission in wounded keratinocytes was also dampened upon MMP12 depletion. Both FN and sAgrin mediated force transmission were unable to rescue the reduced pMLC intensity in MMP12 depleted cells at 30 min post-wounding (Fig. 5k). Together, these results uncover that a deficit of actomyosin tension in MMP12 depleted cells abrogated their mechanoperceiving ability normally conferred by Agrin following skin injury. Therefore, the Agrin mechanotransduction in accelerating the migration of keratinocytes and wound closure is largely dependent on MMP12 in both functional and mechanistic aspects.

**Agrin fails to heal wounds in MMP12 deficient mouse skin.** To test whether sAgrin bestows a mechanically competent wound healing environment by engaging MMP12 in vivo, we depleted MMP12 by stealth siRNAs that efficiently suppressed cutaneous MMP12 levels at days 0 and 10 post-wounding (Fig. S10a, b). MMP12 suppression delayed wound healing rates when compared to those treated with a control siRNA (Fig. 6a, b). Additionally, the accelerated healing rates and the degree of re-epithelization triggered by sAgrin were dramatically suppressed upon MMP12 depletion within 10 days post-injury (Fig. 6a, b). We further noted that increased pMLC expression in the wound edges and beds of control and sAgrin treated ones were significantly attenuated in MMP12 depleted skin sections (Fig. 6c). In addition, sAgrin induced mature and mixed collagen fiber deposition was significantly diminished upon MMP12 depletion at the wound beds (Fig. 6d). Thus, MMP12 depletion renders a mechanically incompetent environment that is majorly deprived of ECM replenishment and acto-myosin tension, which is no longer responsive to sAgrin. Reduced angiogenesis (fewer blood vessels) was also observed within the wound beds of MMP12 depleted mouse skin (Fig. 6e). As such, sAgrin failed to induce robust CD-31 expressing blood vessels when MMP12 was suppressed in the mouse skin (Fig. 6e). Together, these observations reveal that both endogenous Agrin and exogenous sAgrin

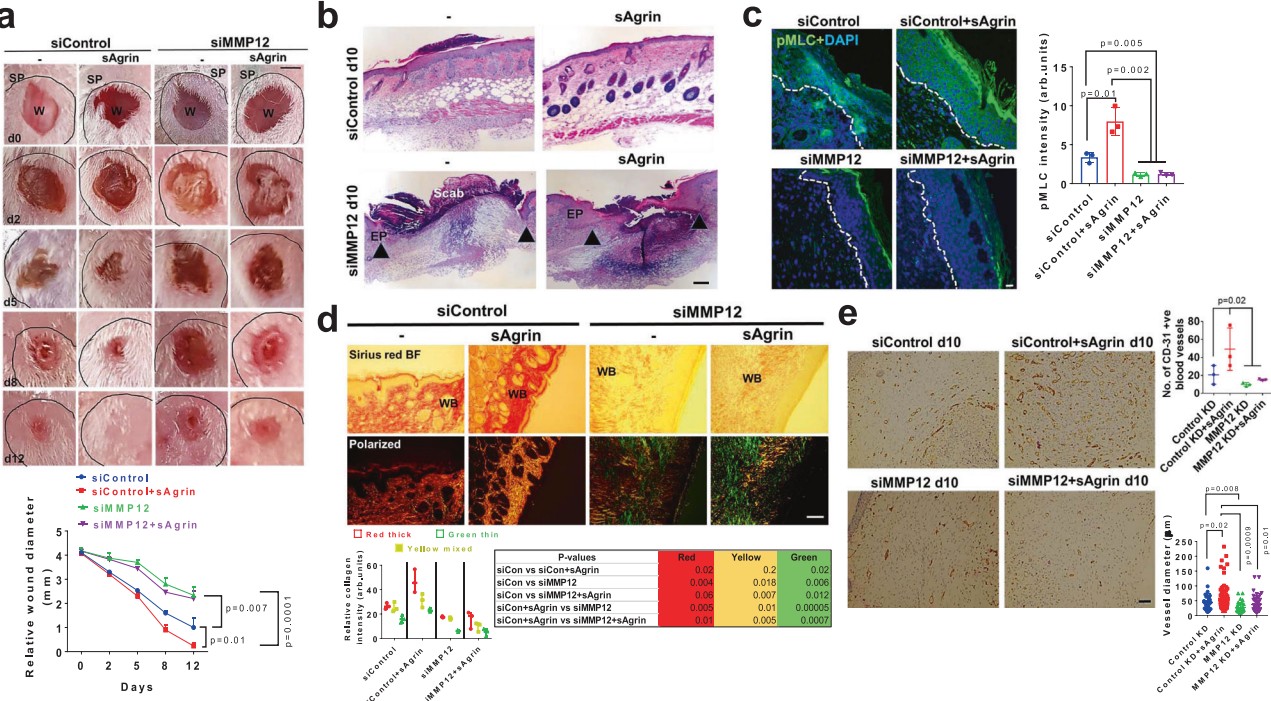

**Fig. 6 MMP12 is required for sAgrin-induced wound healing in vivo. a** Representative photographs of control or MMP12 silenced splinted mouse skin wounds at indicated days either untreated or treated with 200 µg of sAgrin. Scale: 1 mm. The wound diameter at indicated days is quantified as mean +/− s.e.m. (n = 3 mice per group, two-way ANOVA, *p = 0.007, *p = 0.01, ***p = 0.0001, respectively). SP indicates splint. **b** Representative Haematoxylin and Eosin stained sections of mouse skin receiving the combinations of siRNA treatments as in panel (**a**) at day 10 post-wounding. Region encoded by black arrows represents the unhealed area (n = 3 sections from three mice per condition, repeated twice). Scale: 100 µm. **c** Representative confocal images showing the pMLC staining in control and MMP12 depleted mouse skin in the absence or presence of sAgrin treatments. Scale: 10 µm. The pMLC intensity was quantified as mean +/− s.d. (n = 3 mice sections per group, two-tailed Students 't' test, *p = 0.01, **p = 0.005 and 0.002, respectively). **d** Picrosirius red staining images of indicated mouse skin sections at day 10 post-wounding. Representative bright-field and polarized views are presented. Scale: 100 µm. WB denotes wound bed. The collagen fibers distribution was presented as mean +/− s.d. (n = 3 sections per group, two-tailed Students 't' test, p values presented in the table).
**e** Assessment of angiogenesis by CD-31 immunohistochemistry in control and MMP12 deficient mouse skin wound beds in the presence or absence of sAgrin treatment at 10 days post-injury. The number of blood vessels and their diameter are represented as mean +/− s.d. (n = 3 mice per group, at-least 30 blood vessels from each image from three mice per group were analyzed, One-way ANOVA, *p = 0.02, two-tailed Students t-tests, *p = 0.02, p = 0.01, p = 0.008, **p = 0.0009, respectively). Scale: 100 µm.

requires MMP12 to generate a mechanically competent pro-angiogenic wound healing niche in vivo.

**Agrin as bio-additive in hydrogel material accelerates wound healing**. An Agrin-primed environment in wound closure prompted us to test whether sAgrin, per se, accelerated in vitro and in vivo wound healing which may have clinical relevance as a wound healing biomaterial. The following lines of evidence may support the wound healing relevance of sAgrin. First, sAgrin rescued most of mechanotension deficits and provided pro-migratory benefits to Agrin deficient and proficient keratinocytes in the context of wound healing, as well as in liver cancer cells[18,19]. Second, sAgrin bestowed significant angiogenic capability to endothelial cells[20]. Therefore, we generated sAgrin recombinant protein containing 'z' insert by gel-filtration (Fig. S11a). In addition to the fact that sAgrin supported the proliferation of primary mouse keratinocytes as shown by BrdU incorporation assay (Fig. S11b), we noted accelerated migratory rates when keratinocytes were treated with increasing concentrations of sAgrin (Fig. S11c). Further, sAgrin treatment also enhanced the K17 expressing leader keratinocyte migration in vitro, and ex vivo keratinocyte outgrowth from mouse skin explants (Fig. S11d, e). In Agrin depleted cells, force transmission via sAgrin (but not FN) effectively enhanced MMP12 recruitment in pMLC activated regions (Fig. S11f).

Finally, using two independent full-thickness mouse punch-biopsy (both non-splinted and splinted) wound healing models[33,53], we assessed the wound healing attributes of sAgrin versus control proteins when supplied as bio-additive in two different hydrogels for topical application. In the first paradigm, sAgrin or a control protein (BSA) was incorporated in Vaseline, as inert hydrogel material for topical delivery to the wounded skin following a non-splinted healing model. We determined the healing efficacy promoted by different concentrations of sAgrin incorporated in Vaseline. As shown, sAgrin significantly accelerated the healing rates of mouse punch wounds in a dose-dependent fashion in a non-splinted wound healing model (Fig. S11g). Subsequently, we tested a single concentration of sAgrin (200μg) dissolved in Pluronic F-127 as a thermoresponsive hydrogel, extensively used for topical delivery for monitoring the healing process[54]. Likewise, when compared to the control (BSA) topical hydrogel preparations, sAgrin fostered hydrogel treatment significantly boosted the rates of wound healing in vivo (Fig. 7a). The improved healing rate was supported by an enhancement of Ki67 expressing proliferating cells at the base of the migrating epidermis and wound edges of sAgrin treated mouse skin sections (Fig. S11h).

In addition, we tested the efficacy of sAgrin in comparison to rat-tail collagen-I and BSA as bio-additives using splinted wound healing mouse models. The incorporation of 200ug sAgrin within Pluronic F127 significantly accelerated wound healing when compared to similar concentrations of collagen or BSA within the observed 12 days post-injury (Fig. 7b). These results show that sAgrin as a bio-additive may accelerate wound healing response irrespective of the hydrogel material used for topical delivery in two independent mouse models for wound healing. Furthermore, a significant extension of the migratory epithelial tongue was observed upon sAgrin treatment (Fig. 7c). In splinted conditions at day 10 post-wounding, sAgrin treated animals predominantly represented healed skin with the emergence of regenerated hair follicles in the vicinity of the wound area in comparison to BSA or Collagen treated groups (Fig. 7c). This was accompanied by robust K17 expressing keratinocyte outgrowth, MMP12 expression and pMLC activation observed both at wound edges and the wound beds of mouse skin receiving sAgrin based hydrogels

during the early phases (day 2) of wound healing in non-splinted models (Fig. 7d–f). As wound healing required a longer time-frame under splinted conditions, we deciphered a similar increment of K17 expressing cells, MMP12 expression, and pMLC activity at day 7 post-injury in sAgrin treated animals in comparison to BSA or Collagen treated counterparts (Fig. 7d–f). Since ECM replenishment within the wound bed guides the overall healing process, sAgrin treated wound beds exhibited higher deposition of mature and mixed collagen fibers when compared to BSA and Collagen treated groups (Fig. 7g). Similar enhancement of collagen deposition by sAgrin was observed within the wound beds at day 7 in non-splinted conditions (Fig. S11i). Therefore, these shreds of data suggest that sAgin promotes optimal collagen deposition in the wound bed to sustain the degree of stiffness within the wound healing niche.

Prompt expression of inflammatory cytokines early during the wound healing process is known to initiate an efficient skin healing program[55]. Hence, we next explored whether sAgrin impacts key cytokines and chemokines to modulate inflammatory responses associated with wound healing. We profiled the mRNA expression of several cytokines and chemokines known to be induced within 48 h after injury to the mouse skin[56]. Among them, TGF-β1, VEGF-A, MCP-1, MIP-1β, and MIP-2 were significantly induced in the mouse skin wounds within 48 h post-injury that received sAgrin treatment in comparison to BSA or Collagen treated groups (Fig. S12a). Indeed, augmented protein expression(s) of TGF-β1, VEGF-A and MCP-1 were confirmed in sAgrin treated groups, when compared to BSA and Collagen treated mouse skin wound tissues (Fig. S12b). Interestingly, TGF-β1 and VEGF-A have been previously identified as promoters of keratinocyte re-epithelization, angiogenesis, tissue remodeling, and healing[57,58], while MCP-1 is required for re-epithelization of skin wounds[59]. Therefore, timely engagement of a highly selective set of pro-inflammatory proteins by sAgrin is consistent with an accelerated wound healing program.

We further established whether improved healing rates attributed by sAgrin occurs via sustaining angiogenesis in the wound beds. A steady enhancement in angiogenesis with an increased number of blood vessels and their respective diameters was observed in the Agrin-treated mouse skin wound beds on day 6 in non-splinted models (Fig. 7h). Compared to that of BSA or Collagen, an increased number of blood vessels and their diameters were also observed in sAgrin treated mouse skin wound beds at day 12 post-injury in splinted models (Fig. 7h). As such, our in vivo data suggests that Agrin may boost the interaction of dermal fibroblasts and endothelial cells, thereby enhancing angiogenic activity within the wound bed. To test this in vitro, we depleted Agrin in human dermal fibroblasts (GFP-tagged BJ) and human dermal microvascular endothelial cells (cell-tracker blue labeled HDMEC) and subjected them to a 3D co-culture sprouting angiogenesis assay (Fig. S13a). Notably, the control spheroid co-cultures generated robust sprouting which was significantly diminished by suppressing Agrin in endothelial cells and fibroblasts individually (Fig. S13b, second and third panels, respectively). Interestingly, depleting Agrin simultaneously in the fibroblasts and endothelial cells resulted in stronger abrogation of sprouting angiogenesis (Fig. S13b). Together, these results indicate that Agrin supports a pro-angiogenic environment in the wound bed by fostering interactions between the dermal fibroblasts and endothelial cells. Therefore, in addition to laying a solid platform for usage as biomaterial promoting skin tissue repair, our study suggests that topical delivery of sAgrin accelerated in vivo wound healing by augmenting a mechanotransducing environment that sustained optimal ECM deposition and angiogenesis to promote keratinocyte proliferation and migration for wound healing.

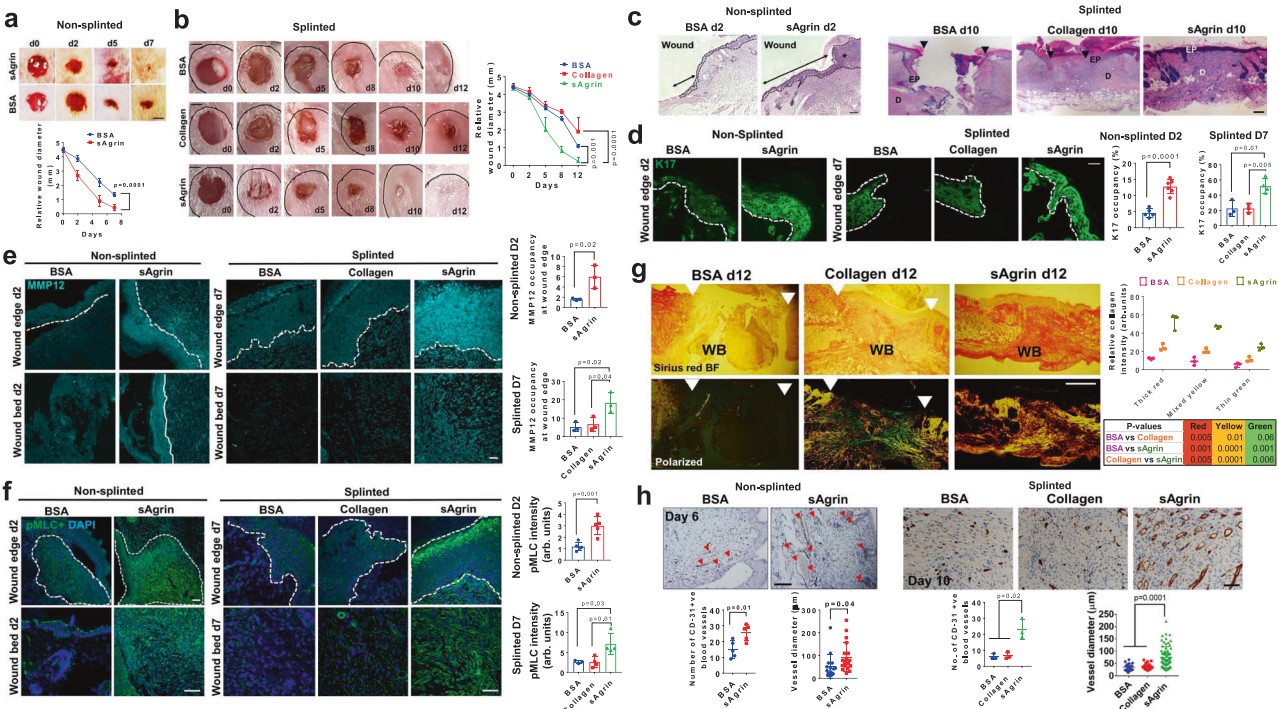

**Fig. 7 Accelerated skin wound healing by sAgrin-based topical biomaterial.** Representative photographs showing the wound areas in mice receiving the indicated treatments at respective days under non-splinted (**a**) or splinted (**b**) conditions. Scale: 1 mm. Mean wound area +/− s.e.m. is shown (n = 4 mice per group, Two-way ANOVA, p = 0.0001, p = 0.001, p = 0.0001, respectively). **c** Hematoxylin and Eosin stained bright-field images of control and sAgrin wound sections at day 2 (n = 3 sections from three mice per group, two experiments). Asterisk (*) denotes the migrating keratinocytes. The black arrow (non-splinted) shows the migrating keratinocytes. The black arrows (splinted) present the area left unhealed at day 10 post-wounding. Scale: 100 μm. **d** Confocal images showing K17 occupancy in the mouse skin wounds receiving BSA, Collagen, or sAgrin treatments in non-splinted (at day 2) or splinted conditions (at day 7). Relative K17 intensity is presented as mean +/− s.d. (n = at-least three sections analyzed from 3 mice per group, two-tailed Students 't' test, *p = 0.01, **p = 0.005, ***p = 0.0001, respectively). Representative confocal images showing MMP12 (**e**) and pMLC (**f**) at the wound edges and beds of mouse skin receiving treatments at day 2 (non-splinted) or day 7 (splinted) post-injury. Relative staining intensities are presented as mean +/− s.d. from three sections from three mice (two-tailed Students 't' test, *p = 0.02, p = 0.04, p = 0.03, p = 0.01, and ***p = 0.001, respectively). Scale: 100 μm. The dashed white line separates the epidermis from dermis regions. **g** Picrosirius red-stained images of BSA, Collagen, and sAgrin treated mouse skin sections (splinted model) at day 12 post-wounding. Scale: 100 μm. WB denotes wound bed. White arrows represent the unhealed area. The collagen fibers were quantified as mean +/− s.d. (n = 3 sections from three mice per group, two-tailed Students 't' test). **h** CD-31 immunohistochemistry in BSA, Collagen, and sAgrin treated mice skin at indicated days. Red arrows denote blood vessels. The number of blood vessels and their diameter are represented as mean +/− s.d. (n = at-least 3 sections analyzed from three mice per group, two-tailed Students 't' test, *p = 0.01, *p = 0.04, One-way ANOVA *p = 0.02, and ***p = 0.0001, respectively). Scale: 100 μm (non-splinted) and 50 μm (splinted).

## Discussion

Upon tissue injury, composite changes in the disorganized ECM and its surrounding tissues that integrate a collage of mechanical forces, tissue compression, bulk stress, and collective migration of cells are required to initiate and support the healing program[60,61]. Notably, concerted replenishment of the wounded ECM sensitizes the wound environment towards adverse physical stresses leading to a productive healing response and limits hypertrophic scarring[5]. Comprehensive lines of evidence presented here uncover several important findings that advance our understanding of mechanobiology in wound healing and offer potential significance to tissue repair strategies. Firstly, Agrin represents a vital ECM proteoglycan whose expression is triggered in the wounded skin tissue and this Agrin-enriched microenvironment tunes the mechanical landscape for productive wound healing (Fig. 8). Secondly, Agrin guides collective keratinocytes migration over the wounded sites by sensitizing them towards different forms of physical parameters such as bulk ECM rigidity, mechanical force, and geometrical constraints. Thirdly, the mechanoperception ability conferred by Agrin robustly overhauls the cytoskeletal architecture following wound injury and is largely dependent on MMP12 activation. Thus, Agrin mechanotransduction via MMP12 activation provides an integrated

mechanism that empowers wounded skin cells to retaliate adverse mechanical stress and accelerate their migration. Fourthly, appreciating the importance of a mechanically competent Agrin enriched wound healing environment, our results additionally reveal that a recombinant soluble Agrin fragment may further accelerate the wound healing process. Hence, appropriate utilization of sAgrin as a bio-additive wound healing material may offer potential clinical value for wounds that are naturally reluctant to heal, as presented by diabetic patients. Furthermore, our study would pave the way to investigate whether Agrin levels are decreased in the skin tissues of diabetic patients and those infected with bacterial infections, presenting a basis for chronic delayed healing of wounds in these patients. Since sAgrin as bio-additive may provide a considerable degree of mechanical stimuli enabling collagen remodeling and re-epithelization, we suggest that sAgrin therapy may have a potential for diabetic wound closure. Future studies involving sAgrin as a wound healing bio-additive in diabetic mouse wound healing models will also reveal the potential in future translational research among diabetic patients. In circumspection of the fact that sAgrin restores the collagen (and possibly other ECM proteins) replenishment while preserving the mechano-architecture of the wound environment, we anticipate a prominent wound closure and tissue

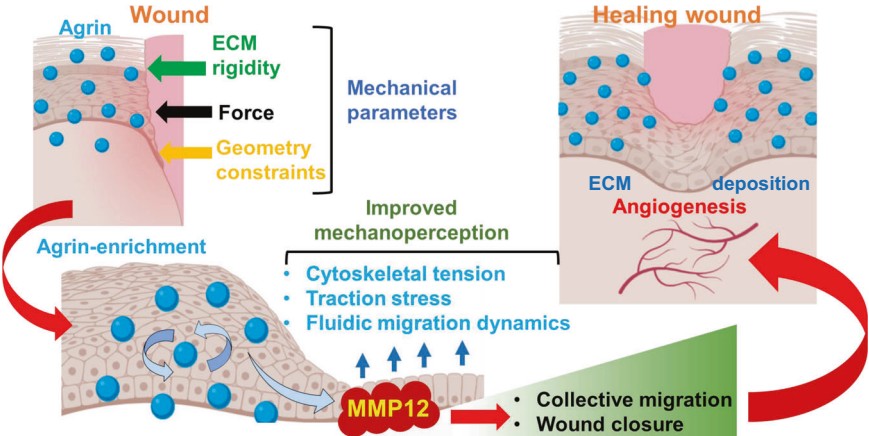

**Fig. 8 A working model explaining that wound injury triggered Agrin microenvironment favors a productive healing program.** Briefly, Agrin sensitizes the wounded cells towards ECM rigidity, force recognition, and geometric constraints accounting for improved traction stress, elasticity, and cytoskeletal tension to promote enhanced fluid-like dynamic collective migration (blue arrows) over the wounded sites. Further, Agrin activates MMP12 as its downstream effector to upgrade the mechanoperception of keratinocytes and actomyosin integrity during keratinocyte migration. The cumulative outcome of an Agrin-driven mechanically competent wound healing micro-environment yields higher collective migration and wound closure via facilitating re-epithelization, optimal ECM deposition, and angiogenesis.

repair role associated with Agrin rather than in de novo skin tissue regeneration.

Epithelial cells experience a wide array of physical stresses that include ECM rigidity, topographic changes in cell shape and geometry, lack of adhesion, and application of mechanical forces[61]. Each of these parameters dictates cell behavior in a wound environment, however, the nature of ECM proteins that enable wounded cells to respond to such physical parameters is less known. Driving this conviction, Agrin, in part, sensitizes keratinocytes to enforce collective migration in response to several physical parameters including bulk ECM rigidity, shift cytoskeletal tension upwards in geometrically constrained architectures, and act as localized mechanotransducer when extrinsic forces are applied to wounded cells (Fig. 8). Similarly, Syndecan-4, a related Heparan Sulfate protein, is required for accelerated wound healing via cytoskeletal rearrangements, and recently attributed as a mechanotransducer of localized force that promoted cell stiffening via integrin-Rho-A axis[13,47,62]. Therefore, whether Agrin collaborates with Syndecans to facilitate the stiffening of the wound environment via integrins or related mechanosignalling poses an interesting path for future investigations. Nonetheless, since FN engagement of integrins failed to restore actomyosin cables in the absence of Agrin upon extrinsic force, the magnitude of cytoskeletal overhauling and enhanced mechanoperception in response to a wound microenvironment appears to be unique for Agrin mechanotransduction. Consistent with the notion that collective cell migration towards a stiffened ECM generates higher traction forces[63], the potential of Agrin as an ECM player is of significance. While Agrin depletion independently inhibited 2D scratch wound healing in keratinocytes and dermal fibroblasts, 3D stiffness-dependent keratinocyte fibroblast co-culture migration assays revealed that dermal fibroblasts sourced Agrin is insufficient to restore the migration of Agrin depleted keratinocytes on stiff substrates. Also, the knockdown of Agrin in the dermal fibroblasts did not impact the migratory rates of overlaid keratinocytes with intact Agrin on stiff substrates. Given that non-wounded and wounded keratinocytes expressed higher levels of Agrin than in dermal fibroblasts, Agrin expressed by keratinocytes acts as a master regulator coordinating migration over wounded regions upon sensing bulk ECM rigidity.

The dynamic ECM reorganization of an Agrin-enriched environment-induced shortly after wounding remains yet to be fully established. Our transcriptomic data reveals an overall loss of collagens and ECM structural components in Agrin depleted keratinocytes. Given that Agrin depletion delays wound healing, it is evident that Agrin confers comprehensive ECM reorganization signals via MMP12 (and other yet-to-be-identified proteins) to sustain wound healing. This is substantiated by the observation that sAgrin within the ECM restores in vitro and ex vivo migratory response within keratinocytes, which was blunted by the depletion or inhibition of MMP12, suggesting that MMP12 is essential for sAgrin to execute its functional outcome. For instance, keratinocyte outgrowth from mouse skin explants on collagen gels that closely mimic the stiffness of underlying dermis[64], was robustly increased when sAgrin was incorporated with the subsequent activation of MMP12. Moreover, in addition to manipulating the tissue mechanics offering better resilience to mechanical injury, the accelerated healing effects illustrated by sAgrin hydrogel treatment in vivo through MMP12 activity likely involves dynamic ECM reorganization, as suggested by increased collagen deposition. Yet, how the wounded ECM is refurbished by an upgraded Agrin-MMP12 mechanotension on the skin cells remains unclear. Since MMP12 activity has a limited association with cytoskeletal tension and/or mechanotransduction, further studies are required to understand the ECM components, which when specifically processed by MMP12 favor a mechanically competent wound healing niche and at the same time reduce the risk of scarring from excessive mechanosignalling[65].

Several studies have revealed that resuspension of siRNAs with topical formulations (such as Pluronic F127 or Aquaphore gels) may effectively inhibit target gene expression in the skin layers[30,66]. For instance, siRNAs against YAP and TAZ in the mouse skin have been associated with a delayed wound healing response[66]. Likewise, our study employed several stealth siRNAs which were very effective in knocking down Agrin and MMP12 in the skin layers (Figs. 2 and 6). Repeated topical siRNA dosage effectively inhibited the Agrin-MMP12 protein expression up to day 10 post-wounding and impacted the course of healing (Figs. S2c and S10b). We envision that the conjugation of these Agrin-MMP12 siRNAs with nuclei-acid nanoparticles may improve their specificity and stability to emerge as future topical delivery agents to study skin wound healing disorders. Further research involving the conjugation of Agrin or MMP12 siRNAs with nucleic-acid-based nanoparticles may be commissioned to

broaden the translational impact of targeting key proteins in skin wound healing disorders.

In sum, we reveal that injury-triggered Agrin enrichment integrates a broad range of mechanical stimuli compelling enhanced mechanoperception via MMP12 activity in keratinocytes that ultimately favor the wound healing tissue microenvironment. Given the potential clinical value of scaffolds promoting tissue regeneration post-injury[67], our findings lay a solid foundation for designing sAgrin-based scaffolds that may assimilate optimal mechanobiological parameters for efficient wound healing.

## Methods

**Antibodies, siRNAs, and reagents**. The following antibodies were used in the study: Mouse monoclonal anti-Agrin (D-2), Santa Cruz Biotechnology, Cat#sc-374117; RRID: AB_10947251 (WB dilution: 1:250; IF-IHC: I:50); Human Agrin clone PIF12 (in house generated, WB; 1:250); Rabbit polyclonal Agrin antibody, Novus Biologicals, Cat# NBP1-90209, IHC 1:50; Mouse monoclonal anti-Integrin β1 Clone P5D2, Abcam, Cat#ab24693; RRID: AB_448230 (IF dilution 1:50); Anti-Integrin Beta1, activated, Clone HUTS-4, Sigma, Cat# MAB2079Z, RRID:AB_2233964 IF dilution 1:50; Mouse monoclonal anti-β-Actin (C4), Santa Cruz Biotechnology, Cat#sc-47778; RRID: AB_2714189 WB dilution 1:1000; Rabbit polyclonal anti-GAPDH (FL-335), Santa Cruz Biotechnology, Cat#sc-25778 WB Dilution 1:1000; RRID: AB_10167668; Phospho-Myosin Light Chain 2 (Ser19), Cell Signaling Technologies, Antibody #3671, RRID:AB_330248 IF dilution 1:50, Anti-Myosin light chain (phospho-S20) antibody (ab2480), RRID:AB_303094 WB Dilution 1:1000; Myosin Light Chain 2 Antibody #3672, RRID: AB_330278 WB Dilution 1:1000; MMP-12 Antibody clone G-2, Cat# sc-390863 WB Dilution 1:500; Anti-MMP12 antibody, Abcam, Cat# ab137444 IF-IHC 1:50; MMP12 Polyclonal Antibody, Thermo Fisher Scientific, Cat# PA5-27254, RRID: AB_2544730; Rabbit polyclonal anti-CD31, Abcam, Cat#ab28364; RRID: AB_726362 IF-IHC 1:50; Goat anti-rabbit IgG-HRP, Santa Cruz Biotechnology, Cat#sc-2030; RRID: AB_631747 WB dilution 1:2000; Goat anti-mouse IgG-HRP, Santa Cruz Biotechnology, Cat#sc-2005; RRID: AB_631736; Goat anti-mouse IgM-HRP, Santa Cruz Biotechnology, Cat#sc-2973; RRID: AB_650513 WB dilution 1:2000; Goat anti-Rabbit IgG (H + L) Alexa Fluor 488 Invitrogen, Cat#A11034 IF dilution 1:500; Goat anti-mouse IgM Alexa Flour 594, Invitrogen, Cat#A21044 (IF dilution 1:500), Anti-BrdU, Abcam, Cat# aab8152 (IF dilution: 1:50). The following siRNAs were used in this study: human Agrin Stealth siRNAs (Set of 3) HSS139721, HSS180123, HSS180124; mouse Agrin siRNAs (Set of 3) MSS201833, MSS201834, MSS201835 and human GPC1 Cat# 10845 (Thermo Fisher Scientific), human Agrin smartpool (Cat# L-031716-00-0050), human MMP12 (Cat# L-005954-00-0050) from Dharmacon. The specific siRNA sequences is reported in Supplementary Table 3. Lipofectamine RNAimax (Invitrogen) was used for siRNA transfections following the manufacturer's recommended guidelines. Fibronectin was obtained from Gibco, and Advanced Biomatrix. Rat tail collagen type 1 was from Corning. Alexa-488 conjugated F-actin phalloidin was from Thermo Fisher Scientific. Blebbistatin and Pluronic F127 was obtained from Sigma and MMP408 from Merck Millipore, respectively. Aquaphore was obtained from Beiersdorf AG while Vaseline was from Unilever. Dimethylsulfoxide was from Kanto Chemical, co., Inc., Cat#10378-00. CellTiter 96® AQueous One Solution Cell Proliferation Assay (MTS), Promega Cat# G3580, was used for proliferation assays. 5-bromo-2′-deoxyuridine-BrdU (#Cat ab142567), Abcam.

**Mice**. We used six to eight to week old female experimental ICR (nomenclature: IcrTac:ICR) mice purchased from InVivos for punch-biopsy wound healing in vivo assays. The mice were housed under standard conditions of 21 °C and a 12 h light-dark cycle with free access to food. In the ex-vivo mouse skin explant assay, random ratios of male and female new born ICR mouse pups were used. The experiments were performed taking into consideration an ethical and reductionist approach of animal usage. Animal studies were not performed in a blinded fashion. The animals were randomly assigned to different experimental groups. The number of animals used for each experiment is indicated in the legend as $n = x$ mice per group. All animal experiments performed were done in accordance with experimental protocols reviewed and approved by the Biological Resource Center (BRC), Agency for Science Technology and Research (A*STAR) under strict compliance to the Institutional Animal Care and Use Committee (IACUC) guidelines for ethical use of animal models in biological research.

**Cell lines and human skin explants**. Human epidermal keratinocyte cell line HaCaT was maintained in Dulbecco's modified Eagle's Medium (DMEM) (Gibco) containing 10% fetal bovine serum (FBS) with Penicillin and Streptomycin (Gibco Cat#15140148) antibiotics. The human foreskin normal fibroblasts (BJ) obtained from ATCC (ATCC® CRL-2522™) were passaged in DMEM (Gibco) with antibiotics. Normal primary adult epidermal keratinocytes (HEK) purchased from CELL Applications, Inc., (Cat# C-12003) were maintained in manufacturer-provided keratinocyte growth medium. Primary keratinocytes from C57BL mouse

strain were purchased from CellBiologics (Cat #C57-6066K) and maintained as per manufacturer-recommended Epithelial cell growth medium (Cat#M6621, Cell-Biologics). The mouse primary dermal fibroblasts were isolated from C57BL mouse strain (Cat#m-GFP-6067), and cultured in a complete fibroblast medium (Cell-Biologics, Cat#M2267). Epiderm full-thickness (EPIDERM-FT™) human skin explants on a basement membrane bearing 3 mm punch wounds were purchased from MatTek Lifesciences and maintained using the company-supplied medium. All cells were propagated at standard culture conditions of 37 °C and 5% $CO_2$.

**RNA interference and knockdown in vitro and in vivo**. The siRNAs were dissolved as per the manufacturer's recommended buffers. For in vitro studies, 50 nM of control or targeted siRNA was mixed with 5 µl of Lipofectamine RNAimax (Thermo Fisher Scientific), and incubated for 30 min at room temperature. The siRNA: lipofectamine mixture was added to 250 µl of reduced growth factor medium. The knockdown was verified by RT-PCR or Western blot after 72 h. For in vivo and ex vivo RNAi, 75 nM of stealth siRNAs were mixed with 8 µl of Lipofectamine RNAiMAX (Thermo Fisher Scientific) for 30 min at room temperature. Subsequently, the siRNA mix was slowly incorporated into a 1:1 mixture of Aquaphore gel in Phosphate buffered saline (PBS) for topical delivery. For treatment with skin explants, the siRNA lipofectamine mix was added to Aquaphore as 1:1 mixture at day 0 post-wounding and incubated for the indicated number of days. For efficient knockdown, the topical siRNA ointment was re-applied on day 4 post-wounding. For the explants that were rescued with sAgrin, an Aquaphore mix containing 20µg protein was added on days 2, 4, and 6 post-wounding. See Supplementary Table 3 for siRNA sequences used.

**Recombinant sAgrin expression and purification**. The C-terminus fragment containing 8 amino acid 'Z8' insert and LG3 domain was cloned into a pET28 vector containing His-ta and expressed in *Escherichia coli* (strain BL21-DE3). The bacteria were transformed with the plasmid. A single colony was inoculated in TB medium containing antibiotics and induced with IPTG for 16 h at 15 °C. The cells were harvested by centrifugation and cell pellets were lysed in lysis buffer followed by sonication. The resultant supernatant containing sAgrin fragment was purified using gel filtration chromatography via Superdex75 columns in imidazole buffer, and sterilized by passing through a 22 µm low-protein binding filter and was dissolved in PBS solution.

**Punch-biopsy wound healing models**. The mice were intraperitoneally (i.p.) anesthezised using Ketamine 100 mg/kg and Xylazin 10 mg/kg diluted in 100 µl of saline solution. The fur was shaved from the base of the neck towards the back on the entire shoulder region. The skin was wiped with an alcohol swab and 10% povidone-iodine (Betadine) antiseptic solution. Circular symmetrical 4 mm wounds were inflicted on either side of the midline in the shoulder region using a sterile 4 mm punch biopsy needle (Integra Miltex, Integra York, P.A., Inc.). The wound skin was carefully removed using a scalpel and a pair of scissors to generate a full-thickness wound that was left open for topical administrations during the analyzed time periods. For splinted wound healing models, a 10 mm transparent donut-shaped nylon sheet (Grace Bio-Laboratories, Bend, OR) was placed around the 4 mm wound. The splint was placed with the wound at the center, glued to the skin by adhesive (Krazy Glue®; Elmer's Inc.), and covered by Tegaderm dressing. An adhesive-based non-sutured splinting of nylon sheet to the mouse skin was employed as per previously established protocol[68]. As such, this method relieved pain to animals caused due to suturing and restored the mechanical and histological characteristics similar to sutured nylon splints. The resulting splints were held intact by periodic glue addition (every 2 days) during the course of the study. The wounds were treated with different formulations based on either petroleum jelly (Vaseline), a thermoresponsive hydrogel (Pluronic F127-Sigma) or a commercially available skin ointment (Aquaphore- Beiersdorf, Inc.). For Vaseline based ointment cream preparation, 10 mg per cm² vaseline ointment was mixed with filtered PBS solutions containing the indicated amounts of Agrin (100 µg, 200 µg or 500 µg) and 100 µg BSA (w/w) were used and applied topically every two days. For siRNA-mediated knockdown experiments, 75 nM of indicated siRNAs were incubated with lipofectamine in PBS. Subsequently, a 1:1 mixture of PBS (containing the siRNAs): Aquaphore was applied topically on the wound region every two days. A topical formulation comprising of 20% (w/w) Pluronic F127 (Sigma) was used to dissolve in sterile and 0.22 µ filtered solution containing a final concentration of 200 µg/ml sAgrin or BSA. This mixture was dissolved overnight at 4 °C to ensure that the liquid phase is achieved. The mixture forms a homogenous gel at temperatures above ~20 °C and was dispersed topically to cover the wound. Wound closure was assessed using a Vernier caliper every alternate day. For both models, the animals were caged individually throughout the observed time-points of healing at the institutional animal facility. All wound healing animal experiments were performed in accordance with experimental protocols reviewed and approved by the Biological Resource Center (BRC), Agency for Science Technology and Research (A*STAR) under strict compliance to the Institutional Animal Care and Use Committee (IACUC) guidelines for ethical use of animal models in biological research.

**Mouse skin explant assay**. Three days old ICR strain mouse pups (both male and female) were sacrificed and the hind skin on either side of the midline in the shoulder region was washed in 70% ethanol and excised using scalpel. The excised skin was washed in 70% ethanol and placed (dermis side down) on the pre-made gel substrates of defined stiffness, either in the absence or presence of 20 µg/ml sAgrin. The explants were allowed to adhere for 2 h before adding the keratinocyte culture medium (DMEM) containing 300 µM Ca$^{2+}$,[31]. The keratinocyte outgrowth and the skin tissues were regularly imaged under an Axiovert-200 inverted microscope and were subsequently fixed in 4% paraformaldehyde and immunostained for Keratin 17 antibody.

**Substrate stiffness manipulations and micropatterning**. Tissue culture plates were coated with Col-T-gel (Fischer Scientific) of stiffness ranging from 0.8 (soft) to 30 KPa (stiff) as per manufacturer recommendations and allowed to solidify for 40 min at 37 °C inside the tissue culture incubator. We mixed 0.8 KPa gel (Cat# P720S-component a) or 30 KPa (Cat# P720H-component a) with component b (supplied by the company) to obtain the desired stiffness as per the manufacturer's recommended guidelines. The mixture was layered on the tissue culture plates before adding the cells. Some experiments were performed with cells seeded on the poly-hydrogel plates of defined stiffness: 0.2 and 30 KPa soft and hard silicone polyhydrogels, respectively (CytoSoft, Advanced Biomatrix, Inc.,). The determination of stiffness of silicone substrates was done by the manufacturer as per previously established protocols[69]. Briefly, to prepare silicone gel substrates, 34 × 50 mm microscope cover glasses were spin-coated with a gel pre-polymer (B and C components of Sylgard 184), using a home-built spin-coater. Before the spin-coating, 40 nm carboxylated polystyrene fluorescent beads (Invitrogen, with excitation/emission maxima of 690/720 nm) were added on the glass surface. The gel pre-polymer was cured by baking at 100 °C for 2 h. The gel was then treated with 3-aminopropyl trimethoxysilane for 5 min and incubated for 10 min at room temperature under a suspension of 40 nm beads in a 100 µg/ml solution of 1-Ethyl-3-(3-dimethylaminopropyl) carbodiimide (EDC) in water to covalently link beads to the gel surface. The elastic moduli was measured by tracking beads on the silicone gel substrate as visualized under a wide-field fluorescence microscope. For certain experiments, indicated amounts of sAgrin were incorporated into the 0.8 KPa collagen gel before gelation process. Upon gelation, trypsinized cells or excised mouse skin explants were plated onto the gels of different stiffness either alone or containing sAgrin. The cells/tissues were incubated under standard culture condition with 500 µl recommended culture medium for the indicated days. The crossbow-shaped micropatterns with fibronectin-coated islands were generated by microlithography on a 19.5 × 19.5 mm coverslide from Cytoo, Inc., France. The micropatterns had surface areas of 800 µ$^2$ or 1600 µ$^2$, respectively. Fifty thousand cells after initial siRNA treatment were placed on the micropatterns for 4–6 h before processing them for immunofluorescence. For some experiments, 10 µg/ml sAgrin was added on the slides 18 h prior to the addition of cells.

**3D- stiffness dependent keratinocyte fibroblast co-culture wound healing assay**. A 3D in-vitro wound healing model was created using collagen constructs of defined stiffness by modifying a previously established protocol described in detail[70]. Collagen constructs were made with Col-T-gel (0.8 KPa-soft) or (30 KPa-stiff) as per the manufacturer's recommended protocol containing 300,000 primary mouse dermal fibroblasts (DFs). For some experiments, siControl and siAgrin treated DFs were incorporated within soft or stiff collagen constructs. The constructs bearing fibroblasts were allowed to solidify for 45 min inside the incubator. A second layer of collagen matching the stiffness of the underneath layer was added containing 350,000 of primary mouse keratinocytes (KRTs) and the construct was allowed to solidify for 3–4 h. After removing excess media, the constructs were washed gently with 1× PBS twice. An upside down 2 µ pipette tip was inserted and gently rotated at the center of the constructs to create a circular wound. The collagen and cells were quickly removed from the wound area and replaced with 40 µl of soft or stiff Col-T-gel matching the consistency and set-up of the constructs. The resultant wounded constructs were placed inside the incubator and imaged after 30 min for day 0 time-points. The migratory keratinocytes were imaged till day 5 post-wounding.

**Western blot analysis**. Indicated cells post-manipulation were washed twice with cold PBS and lysed with cold 1% NP-40 lysis buffer supplemented with 1X protease inhibitor cocktail (Roche Applied Biosciences) for 15 mins at 4 °C. The cell lysate was centrifuged at 18,928 × g for 15 min. This was followed by protein estimation using Bradford reagent. Subsequently, 40–50 µl of total protein was mixed with an equal volume of 2X Laemmli sample buffer and heated at 95 °C for 5 mins. This was followed by resolution with SDS–PAGE gel. The resolved proteins were transferred onto nitrocellulose membrane and blocked in 5% skimmed milk reconstituted in 1X PBS containing 0.1% Tween-20, and probed overnight with the respective primary antibody. The membrane was then washed with 1× PBS supplemented with 0.1% Tween-20; three times at 15 min intervals. This was followed by 1 h incubation in conjugated horseradish peroxidase HRP secondary antibody (Santa Cruz Biotechnology). Post-incubation, the blot was again washed three times as above and then overlaid with enhanced chemiluminescence (ECL) substrate (Pierce/Bio-Rad) and visualized on X-ray film by image processor or digitally

by Chemidoc analyzer (Bio-Rad) and analyzed by Image Lab 6.0.1 software. The density of the various bands was quantified using the Image-J 1.52e software.

**Immunofluorescence and confocal microscopy**. Cells were cultured on eight-well chamber slides or coverslips over-night. Cells were then washed twice with PBS and fixed for 15 min with 4% paraformaldehyde. Subsequently, the cells were permeabilized for 15 min with 0.1% Triton X in PBS containing 1 mM Ca$^{+2}$ and 1 mM Mg$^{+2}$ (PBSCM) at room temperature. The permeabilized cells were incubated with indicated antibodies in fluorescent dilution buffer (FDB) for 1–2 h at RT or overnight at 4 °C, followed by five washes with PBSCM and incubation with secondary antibody; Alexa Fluor (Thermo Fisher Scientific) for 1 h at RT. Slides were again washed five times with PBSCM and mounted with Vectashield medium containing DAPI. The stained cells were then imaged by a Zeiss confocal microscope. The images were processed and analyzed by Zen blue (Zen 2 lite) software. Intensity measurements were acquired using tool selection parameter in image analysis within the Zen 2 lite software (Zeiss).

**Histology and Immunofluorescence-histology**. Full-thickness human skin embedded within matrix and mice skin tissues were harvested, embedded, and subjected to routine histology using Hematoxylin and Eosin. For immunostaining, the tissues were fixed for 24 h in 10% neutral buffered formalin solution. Subsequently, the tissues were sectioned using a microtome and subjected to antigen-retrieval at pH 9 or pH 6 (for the indicated antibodies). Primary antibodies were used at a concentration of 1:50 dilution, while the respective Alexa-488 and Alexa-594 secondary antibodies (Thermo Fisher Scientific) were used at a concentration of 1:100 and 1:500, respectively in fluorescent dilution buffer. The slides were mounted in a VECTASHIELD hardset$^{TM}$ antifade mounting medium containing DAPI. Quantification of collagen as a marker for ECM deposition was done by picrosirius red staining of paraffin-embedded mouse skin tissues, as per previously published protocol as described below[71]. For picrosirius red-stained collagen sections, the tissue staining colors from the polarized images were splitted as individual color images using Image J. The green pixels were considered thin, yellow pixels as mixed and red pixels as thick collagen fibers, respectively.

**In vitro scratch assay**. A confluent monolayer of control and Agrin depleted cells in a six-well plate was subjected to a unidirectional scratch using a 20 µl pipette tip. This was followed by washing with 1× PBS at room temperature and incubation in complete culture media at 37 °C and 5% CO2 with or without sAgrin as indicated. Phase-contrast images of the wound area were taken periodically at the indicated time-points. For some wound-healing migration assays, keratinocytes were cultured on either soft or stiff substrates within a stencil barrier (Nalge Nunc., Inc) for 18 h. Rescue with sAgrin (10 or 20 µg/ml) was performed in batches of Agrin or MMP12 silenced cells for 18 h before analyzing their migratory potential. Upon removal of the barrier, the cells were allowed to migrate and imaged using an Axiovert 200 inverted microscope at indicated time-points and acquired using Axiovision SE64 Rel.4.9.1 SP1 software.

**Traction force microscopy**

*Substrate preparation*. Two-part silicone elastomer, DOWSIL CY52-276 (Dow Inc.), is first mixed thoroughly in a 1:1 weight ratio. The mixture is then poured into a 35 mm glass-bottom culture dish (Iwaki 3930-035, Asahi Techno Glass Corporation, Japan) placed on a level surface, to a thickness of ~500 µm. After pre-curing overnight, the silicone-coated culture dish is further baked for 2 h at 80 °C to fully cure. To maximize subsequent microsphere attachment, the silicone film is pre-treated with (3-aminopropyl) triethoxysilane (A3648, Sigma-Aldrich). To silanize, a 1 ml solution containing 96% v/v, 2% v/v APTES, and 1% v/v acetic acid (A6283, Sigma Aldrich) was added to the glass-bottom culture dish, covering the whole silicone film, and left for 10 min to react. The silicone film was then twice rinsed with 96% v/v ethanol and followed by a quick dip in Milli-Q water to remove all the solution: removing solution from the film this way is preferred over blow-drying with nitrogen gas because the latter may introduce dust to the surface which subsequently affects the uniformity of attached microsphere distribution. Following this, the culture-dish is baked at 80 °C for another 2 h to promote siloxane bond formation. For the physisorption of fluorescent latex microspheres onto the silicone film, 6 µl of 100 nm orange fluorescent (540/560) carboxylate-modified microspheres (F8800, ThermoFisher Scientific) was first diluted in 10 ml of Milli-Q water and sonicated for 10 min. The diluted microspheres are then passed through a 0.22 µm filter directly into a 15 ml falcon tube containing 500 µl of 500 mM MES buffer, pH 6.0 (M3671, Sigma Aldrich). The microspheres were then sonicated again for 10 min before being added to cover the APTES silanized silicone film within the culture dish previously prepared. The solution was left for 5 min to allow the microspheres to attach. Then, to remove the solution, the culture dish was carefully and quickly tipped so that all the solution is removed in a single action, as the solution's surface tension would decouple microspheres if allowed to flow back. The microsphere-coupled culture dishes were then baked at 80 °C for 2 h and typically used within a week. To facilitate the attachment of adherent cells, the microsphere coupled silicone films are coated with human blood plasma fibronectin (10838039001, from Sigma Aldrich) with a concentration of 50 µg/ml to a

surface density of $5\,\mu g/cm^2$. The stiffness of silicone substrates used for TFM was determined by atomic force microscopy as described in the section below.

*Traction force imaging.* A Nikon Biostation IMQ was used to live-cell image the HACAT cell-sheet migration and microsphere displacement over 24 h with 10 min intervals, at 37 °C and 5% $CO_2$. Both phase contrast and epifluorescence images were acquired using the internal 20× objective (0.5 NA) and 1.3-megapixel monochrome camera. The light source for the epifluorescence is the Intensilight Hg Pre-Centered Fiber Illuminator and orange microspheres are imaged using a Texas Red filter set. To account for z-drift, 2–3 planes 1 μm apart either side of the focal plane is typically taken in a z-stack. Three independent samples are imaged for each condition, and for each independent sample, 3–4 locations are randomly selected and tracked.

**Cell stiffness measurement by atomic force microscopy (AFM).** The stiffness of the mouse keratinocyte cells was measured using a Nanowizard IV BioAFM system (JPK Instruments, Germany). Confluent mouse kerantinocyte monolayers were cultured on a petri dish in PBS media. The monolayers were scratched across the center of the petri dish using a 100 μl pipette tip and characterized after 4 h. Indentations were performed on randomly selected cells at the wound edge with a polystyrene bead of diameter (~4.5 μm) attached to the end of a cantilever ($k = 0.03$ N/m, Novascan Technologies, Inc., Ames, IA) using a force of 3 nN at 1 Hz. More than 60 cells from triplicate experiments were characterized and averaged to evaluate the Young's modulus for each condition. One representative experiment is shown in the respective figure panels. Young's modulus values were calculated using JPK Data Processing Software 4.2.1 (JPK Instruments, Germany), which employs Hertz's contact model for spherical indenters (diameter 4.5 μm; Poisson's ratio 0.5) fitted to the extend curves.

**Wound assay using microfabricated device for collective TFM.** Prior to seeding the HaCaT cells, the 35 mm culture dish containing the ECM coated silicone film is setup with a block of polyacrylamide hydrogel ($1.2 \times 1.2 \times 0.5$ cm, see supporting information) which will eventually act to create a clean wound in the HaCaT cell sheet. Polyacrylamide hydrogel is used for the fact that it is resistant to protein binding and would therefore prevent cell adhering to it and also prevent the removal of ECM from the silicone surface. After equilibrating the hydrogel block in the culture dish with growth media for 30 min, the dish is replaced with fresh media, and $\sim 7 \times 10^5$ cells are seeded evenly around the hydrogel block. The setup shown in Supplementary Fig. 14a, b is incubated for 24 h before the hydrogel block is removed, and the cells are imaged thereafter. See Supplementary Methods section for detailed information on traction force measurements contained in Supplementary Information.

**Force transmission via ligand conjugated magnetic beads.** The ligands were conjugated to 4.5 μm epoxide paramagnetic beads (Dynabeads M-450 Epoxy, Thermo Fisher Scientific) according to the manufacturer's manual. Briefly, the beads were washed twice in 0.1 M sodium phosphate buffer, pH 7.4. Subsequently, $8 \times 10^7$ beads were conjugated to 20 μg sAgrin, BSA, and FN per in 0.1 M sodium phosphate buffer at pH 7.4 for 16–18 h at 4 °C with gentle rotation throughout. The degree of protein conjugation to the beads was verified by a reducing SDS-PAGE gel analysis using a subset of ligand conjugated beads. The remaining beads were washed and subsequently stored in 0.1% BSA–PBS pH 7.4 at 4 °C. They were subsequently suspended in medium and added on the cells for 30 min at 37 °C. After a brief wash with PBS to remove excess non-adherent beads, the cells were placed under a permanent neodymium magnet (KJ Magnetics, USA) at a distance of 6 mm apart for another 30 min that allowed a vertical tensile force in the magnitudes of ~200 pN on the beads. The cells were then fixed in 4% paraformaldehyde, permeabilized, and processed for immunofluorescence.

**Dermal endothelial cell fibroblast 3D-angiogenesis assay.** One hundred thousand Human Dermal Microvascular endothelial cells (HDMEC) cells were pre-labeled overnight with Cell Tracker Blue CMAC (7-amino-4-chloromethyl coumarin-Invitrogen) and mixed with equal amounts of GFP expressing BJ cells. Both the cells were treated with the indicated siRNA and allowed to form co-culture spheroids as per previously established protocols[20]. Images of sprouting ECs were captured 24 h post-embedding and quantified using Sprout morphology plugin from Fiji (Image J)[20]. Briefly, tube lengths parameters were used to quantify sprout lengths from spheroids.

**MMP12 activity assay.** MMP12 activity was monitored via Gelatin and Casein zymography as per previously published protocols that are described below[72]. Briefly, control or Agrin depleted HaCaT cells were grown till 80–90% confluency in media without FBS. Supernatants collected from each culture dish were spun down at $11,200 \times g$ for 5 min and concentrated using Amicon ultra columns. Twenty microliters of medium was mixed with 2X SDS loading buffer and loaded on 10% gelatin or casein gel. The gels were washed with incubation buffer and subsequently stained with Coomassie Brilliant Blue for 1 h. The gels were de-stained for 30 min before imaging using a Chemidoc imager.

**Quantitative reverse transcription PCR (RT-PCR) and RNA sequencing.** Total RNA was extracted from indicated cells using Qiagen RNeasy mini kit as per the manufacturer's recommended protocol described as follows. The total RNA was then reverse-transcribed using High-Capacity cDNA reverse transcription kit (Applied Biosystems). The generated cDNA (200 ng) was used as a template for the RT-PCR using the SYBR™Green or Taqman[R] (Thermo Fisher Scientific) based master mix and probes. The data were normalized to GAPDH as endogenous controls. The RT-PCR primers were used in the study for the respective target genes are provided in Supplementary Tables 1, 2. For RNA-sequencing, the RNA quality was analyzed on a Bioanalyzer instrument (Agilent) using the Agilent RNA 6000 Pico Kit. A total of 4 μg RNA was used for RNA-Sequencing library preparation with the TruSeq Stranded mRNA Library Prep kit (Illumina) based on the manufacturer's instructions. Amplification of libraries was limited to 7 PCR cycles. Purified libraries were quantified by qPCR (KAPA Library Quantification Kit for Illumina, Roche). The Agilent High Sensitivity DNA Kit was used to assess the fragment lengths of a subset of libraries, before combining the rest of the libraries into one pool and further subjecting to a sequencing run on a NextSeq500 (Illumina). The condition of sequencing was represented by a single read high output run at 75 bp read length. Raw reads from fastq files were aligned to the hg38 genome using STAR 2.6.1d. Bam files were sorted and indexed with samtools. The relative number of reads mapping to each gene was quantified with htseqcount on features from the gtf file gencode.v29.annotation.gtf. The raw counts of reads were then imported into EdgeR in R to perform the differential expression analyses using the Gene-ontology, GSEA, or other analyses, respectively. Gene set enrichment analysis (GSEA) was performed running the GSEA Pre-ranked tool of the GSEA Software 3.0, using the gene sets from the Molecular Signatures Database (MSigDB) v6.2. Enriched Gene Ontology terms and KEGG pathways were identified using Metascape (https://metascape.org/). Network analysis was done using Cytoscape 3.4.0 (https://cytoscape.org/). Genes with a false discovery rate (FDR) below 1% and more than two-fold change in expression between conditions were considered.

**Statistics and reproducibility.** The number of biological and technical repeats for each experiment is indicated in the figure legends. For most in-vitro experiments, three biological repeats are performed unless stated differently in the legend. For in vivo experiments, no animals were excluded from analysis and the sample size was not pre-determined using power analysis. The age and sex of animals are mentioned in the methods section. Randomization is not applied for experiments using cell lines. Data are presented as mean +/− s.d. or s.e.m. (wherever mentioned in the legend). Two-tailed Students 't' test was employed to detect paired comparisons. One-way or two-way ANOVA (Tukey's multiple comparison test) or Multiple t-tests (Benjamini or Holm−Sidak processes) were used to compare multiple groups using GraphPad Prism software. The data was considered statistically significant when $^*p < 0.05$, $^{**}p < 0.005$, $^{***}p < 0.0005$, respectively. The exact $p$ values and the statistical tests used are presented in the figure legends. Data considered insignificant was designated as 'ns'. No prior statistical tests or assumptions were used to determine the sample size of in vitro, ex vivo, and in vivo experiments. Data are represented from one of three biological replicates unless stated otherwise in the respective figure legends. We chose three biological replicates for in-vitro experiments as it adheres to the commonly held practice in biomedical research. For panels 4a−c, e, 5c, d, h, i, representative images are shown from one of three biological repeats. For Fig. 7d, non-splinted panel: BSA group: five images analyzed from three different mouse wounds; sAgrin group: six images analyzed from three mice; Splinted panel: BSA: three images from three mice, Collagen: three images from three mice; sAgrin: four images from three mice, respectively. Figure 7e: Non splinted panel: BSA group: three images from three mice, sAgrin group: three images from three mice, Splinted panel: three images from three mice for all three groups. Figure 7f: Non-splinted panel: five images from three mice for both BSA and sAgrin groups, Splinted panel: BSA: three images from three mice, Collagen group: four images from three mice, sAgrin group: four images from three mice, respectively. Each dot represents one image showing the staining intensity of respective proteins.

**Reporting summary.** Further information on research design is available in the Nature Research Reporting Summary linked to this article.

## Data availability

All the data supporting the findings from this study are available within the article and its supplementary information. The RNA sequencing data comparing the gene expression profile of control and Agrin depleted keratinocytes have been deposited on Gene Expression Omnibus (GEO Accession number: GSE179322). GSEA (https://www.gsea-msigdb.org/gsea/index.jsp) was performed by Molecular Signatures Database (MSigDB v6.2). Any remaining data will be available from the corresponding authors upon reasonable request. Source data are provided with this paper.

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

## Acknowledgements

The study was funded in part by the Additive Manufacturing for Biological Materials (AMBM) Grant, Agency for Science, Technology, and Research (A*STAR, Singapore) (A18A8b0059) to S.C., and IMCB core funds to W.H. We thank Heike Wollman for the RNA-Seq service and the Advanced Molecular Pathology laboratory at IMCB for assistance in immunohistochemistry.

## Author contributions

S.C. and W.H. conceived the project. S.C. designed and performed all the experiments with assistance from D.S., M.O., and K.N. and inputs from W.H. S.C., M.B., M.H.N., and D.S. performed the T.F.M., A.F.M., and P.I.V. experiments with guidance from C.T.L., S.C., P.G., and K.N. performed the RNA-Seq with inputs from E.G. S.C. and C.C.W. performed the mouse wound healing experiments. S.C. analyzed the data, S.C. and W.H. wrote and edited the manuscript. S.C. and W.H. jointly supervised the project. The funding agency had no role in conceptualizing or analyzing the study.

## Competing interests

The authors S.C. and W.H. have filed a patent application concerning portions of this study. All other authors declare no competing interests.
