## [Peer Review File · Nature Communications]

Reviewers' comments:

Reviewer #1 (Remarks to the Author):

The authors' initial hypothesis is that injury causes loss of ECM, which then inhibits how the underlying tissues may respond to bulk extrinsic mechanical stress, termed "mechanoperception". The authors propose to identify key ECM components that would re-establish mechanoperception and subsequent mechanical stimuli within wounded cells, which would help cells adapt to a wound-stressed environment. Specifically, the authors focused on some ECM proteoglycans Perlecan and Glypicans 1-3 that are frequently overexpressed in wounded cells in vivo, as well as Agrin. While Agrin has been investigated less in the wound healing literature, it has been found in various other publications, including by these authors, to actuate tumor mechanotransduction and have a cardio-protective role following myocardial infarction (Chakraborty, Cell Reports, 2017, Chakraborty, Nat Comm, 2015, Bassat Nature 2017).

The authors found that Agrin was enhanced within days 1-10 post punch wound biopsies in mouse skin and also an in vitro scratch wound model, at both mRNA and protein levels. Additionally, they used siRNA to silence Agrin expression and showed that it delayed wound closure both in vitro and ex vivo, and that this could be rescued with sAgrin treatment.

Agrin expression was found to be highly expressed at the wound edges and/or on stiffer substrates during wound closure, indicating that deposition of this ECM protein was important for providing cell migration and enhancing cell sensing to its environment. Agrin expression was also found to be linked to actomyosin and F-actin expression, in which siRNA depleted actin expression and sAgrin rescued it. These findings are not surprising based off various previous research finding that ECM is needed for migration or stiffness sensing, which has already been established well for fibronectin and also found for Agrin in vitro (Kawahara, Plos One, 2014). Interestingly, the authors found that culturing the keratinocytes on "cross-bow" shaped geometric constraints further showed how sAgrin supplement could reverse siRNA inhibition, shown specifically by an increase in F-actin and pMLC recruitment.

Finally, they use bulk-RNA sequencing to identify transcriptomic changes in control keratinocytes vs siRNA inhibited ones. They identified MMP12 as gene/protein most closely tied to Agrin expression and repeated previously mentioned experiments to show that MMP12 knockout impaired healing that not even sAgrin treatment could rescue. Finally, the authors treated mouse wounds with sAgrin to show that it speeds up wound closure rates.

The authors thoroughly interrogated the effects of Agrin, sAgrin treatment, and MMP12 in their experiments, and their conclusions are sound. However, it is important to note that while these experiments were performed thoroughly and of high quality, their novelty might be limited and they also lacked much physiological relevance, which could limit their impact.

In terms of physiological relevance, the bulk RNA analysis was performed on cells were cultured on extremely stiff plastic environments, which is not physiologic. Many of the main conclusions were made from scratch assays, which by definition have to be performed on stiff plastic. It would be better if the authors could conduct these in vitro wound healing studies in a more physiologically relevant, 3D environment, in which the matrix stiffness more closely matched that of native tissue.

Furthermore, authors identified keratinocytes as the primary drivers of wound closure, but they did not use an animal model that relies on re-epithelialization. It is well known that mice have a panniculus carnosus muscle under their skin, leading them to heal mostly by contracting their wounds. As such, most mice heal their wounds by a quick "purse string" like action, unlike that seen in humans or other large animals. It would be better for the authors to use methods such as splinting in order to promote a more human-like wound healing environment that is driven by re-epithelialization by the keratinocytes (Galiano, *Wound Repair and Regeneration* 2004). This would bring more confidence that their findings are actually primarily driven by keratinocytes and also significantly increase the translational impact of their findings.

In terms of novelty and impact of the findings, previous research (Hattori, *The American Journal of Pathology*, 2009) has already identified both MMP9 and MMP13 as critical drivers of wound closure in keratinocytes using a mouse wounding model without splinting. The novelty of Agrin as a potential mechanotransduction regulator is also brought into question from a search from the literature. The authors clearly have experience studying Agrin and its key role in mechanotransduction and force sensing in a variety of other organ systems such as liver (Chakraborty, *Cell Reports*, 2017, Chakraborty, *Nat Comm*, 2015). Agrin has also been found by others to promote myocardial regeneration (Bassat *Nature* 2017). While the role of Agrin and its link to MMP12 in keratinocytes related to dermal healing has not been specifically investigated, others have recently shown that Agrin promotes corneal wound healing (Hou, *Investigative Ophthalmology & Visual Science*, 2020).

Some minor concerns:

The authors claim Agrin increases in mRNA content, but the figure in 1B does not always show that trend. Additionally, no statistics seem to have been calculated.

It seems like the mouse skin implants did not show any significant difference (figure 2g). I would like the authors to comment on why that might be.

Figure panel organization was sometimes hard to follow, especially since the treatment groups were arranged by column and other times by rows.

The authors chose MMP12 since it had the highest significance, but MMP3 and MMP10 also seemed to have interestingly high fold changes that would also be interesting to evaluate. The authors should mention if they tested this or why they did not interrogate that expression.

Reviewer #2 (Remarks to the Author):

The manuscript entitled, "Agrin-Matrix Metalloproteinase-12 axis confers a mechanically competent microenvironment in skin wound healing" is required major revision before considering this manuscript for publication.

Comments:

1. What about the purity Agrin.

2. How was the stiffness was analyzed? By Rheology? This details should be given.

What is the stiffness is needed for wound regeneration.

3. Size of the wound is not mentioned. It is very important.

Any approved protocol is used for the wound creation model.

4. What about angiogenic nature of Agrin. It should be discussed.

5. Collagen quantification is not shown in histological analysis.

6. How this Agrin contribution in the infectious wounds or diabetic wounds. It should be discussed in the manuscript.

7. What about the quantifications of inflammatory cytokines in the wounds after the treatment.
8. I also suggest that collagen gel can be used as one control in the animal studies.

Reviewer #3 (Remarks to the Author):

In this work "Agrin-Matrix Metalloproteinase-12 axis confers a mechanically competent microenvironment in skin wound healing", Sayan Chakraborty and colleagues investigate if and how Agrin promotes cells migration after skin wounding to stimulate process of healing. Their use of animals in vivo, skin ex vivo, and in vitro assays and human explants/cells approaches is a strength of their work. Moreover, the extension of methods including those focused on Traction force microscopy and Force transmission via ligand conjugated magnetic beads sheds new mechanistic insights into how Agrin enhances the mechanoperception of keratinocytes thereby promoting cell migration participating the skin wound healing process. Additionally, the data showed that Agrin has a promising potential as a bio-additive material for accelerated wound healing. Although the coverage of this manuscript is extensive, the results, particular those related to Agrin-Mmp-12 pathway and the mechanisms are rather descriptive. Usage of animal models as Mmp-12KO mice since Agrin KO is lethal would be of value. Additionally primary culture of mouse cells as a major tool followed by human cell lines would strengthen the data.

1. Figure 1a shows the changes in gene-expression in non-wounded vs post-wounded mouse skin (single-cells transcriptomics analysis). The changes in AGRN expression was not profound as for GPS1 or 2. However in in vitro migration model it is AGRN which showed the most substantial changes, please explain/comment on it.

2. Which cells in the skin are the major source of agrin?

It would be important to show how the content of agrin changes in relation to epidermis, dermis (papillary, reticular and dWAT-component) and how the agrin location/abundance is affected by wound healing process?

3. It would be of value in the cell migration (scratch wound model) use primary mouse skin cells: keratinocytes and DFs, since in vivo data are based on mouse model. Moreover, there are the differences in AGRN protein abundance between HaCaT cells (established cell line) and HEK (primary immortalized line) therefore applying primary keratinocyte/DFs cultures from mouse skin would be of value.

4. How the authors separate cells migration from cells proliferation during in vitro assay.

Re-epithelialization process in vivo is composed by migrating epithelial tongue that is pushed from the back by proliferating keratinocytes. However, in in vitro model the cells migrate and proliferate simultaneously.

5. Immunofluorescent detection of agrin in mouse skin should be also confirmed by immunohistochemical detection in non-wounded and post-wounded skin. IHC detection more clearly should show the abundance in protein localization in the case of agrin epidermis vs dermis in relation to days of injury.

6. Using in vivo model of 4mm diameter wound on the back of mice and then applying scrambled or Agrin siRNA in the ointment at the wound site how you protect animals from the licking ointment off the skin?

7. There are no discussion concerning agrin presence in keratinocytes vs dermal fibroblasts and the possible DFs effect on the process?

8. Based on HaCaT cells there are differences in the expression not only for Mmp-12 but also for Mmp-10 and Mmp-1 (that in mice is rather Mmp-13) why the Mmp-12 was chosen? It is not clear.

9. To exclude that Mmp-10 or Mmp-1 (Mmp-13) is involved in the Agrin action the same set of experiments as for mmp-12 should be performed.

10. Authors showed the zymography for Mmp-12 using gelatin as a product (Extended data figure 4:(b) Gelatin degradation assay detecting the catalytic activity of MMP12 in control and Agrin depleted HaCaT cell supernatants.).

Generally gelatin zymography is used for the analysis of Mmp-2 and Mmp-9 activity. To analyze Mmp-12 activity caseine zymography is recommended. This is interesting that using supernatant from cells culture the authors were able to show Mmp-12 gelatinolytic activity.

(BIOTECHNIQUESVOL. 38, NO. 1REVIEW Zymographic techniques for the analysis of matrix metalloproteinases and their inhibitors Patricia A.M. Snoek-van Beurden & Johannes W. Von den Published Online:30 May 2018<https://doi.org/10.2144/05381RV01>)

11. The authors should also perform the experiments on Mmp-12 KO mice and discuss how skin wound healing/re-epithelialization is accomplished in this mouse model. The use of Mmp-12 KO mice is mandatory to state that Agrin via Mmp-12 reveals its mechanotransduction.

12. MTT is not a proliferation measurement assay but cell metabolic activity assay. To measure the proliferation rate ie. BrdU incorporation assay should be performed.

Point-by-point responses to reviewer's comments

We sincerely thank all the reviewers for their valuable and constructive comments. Following their advice, we have made conscientious efforts to improve the physiological impact of our study. A point-by-point response that address each of the previous concerns is shown below. We sincerely hope that the reviewers will find this revised version acceptable for publication.

Reviewer #1 (Remarks to the Author):

1) The authors' initial hypothesis is that injury causes loss of ECM, which then inhibits how the underlying tissues may respond to bulk extrinsic mechanical stress, termed "mechanoperception". The authors propose to identify key ECM components that would re-establish mechanoperception and subsequent mechanical stimuli within wounded cells, which would help cells adapt to a wound-stressed environment. Specifically, the authors focused on some ECM proteoglycans Perlecan and Glypicans 1-3 that are frequently overexpressed in wounded cells in vivo, as well as Agrin. While Agrin has been investigated less in the wound healing literature, it has been found in various other publications, including by these authors, to actuate tumor mechanotransduction and have a cardio-protective role following myocardial infarction (Chakraborty, Cell Reports, 2017, Chakraborty, Nat Comm, 2015, Bassat Nature 2017).

The authors found that Agrin was enhanced within days 1-10 post punch wound biopsies in mouse skin and also an in vitro scratch wound model, at both mRNA and protein levels. Additionally, they used siRNA to silence Agrin expression and showed that it delayed wound closure both in vitro and ex vivo, and that this could be rescued with sAgrin treatment.

We thank this reviewer for the positive comments summarizing the key aspects of our study. As the reviewer correctly points out that our earlier studies elucidated Agrin's mechanotransducing role in the context of the liver tumor environment. However, the role of Agrin as an extrinsic force transducer upon skin injury that upgrades the mechanoperception of keratinocytes is the new highlight of this present manuscript. Specifically, we demonstrated here how Agrin promotes mechanoperception in the wound environment reinforcing efficient repair mechanisms. Despite the fact that numerous proteoglycans control wound healing, a similar mechanotransducing function has not been reported for any other proteoglycan thus far. Moreover, the mechanotransducing effects of Agrin has not been reported in any regenerating tissues including heart, bone or cornea. Therefore, our study unmask a new mechanistic insight of Agrin role to promote skin repair.

2) Agrin expression was found to be highly expressed at the wound edges and/or on stiffer substrates during wound closure, indicating that deposition of this ECM protein was important for providing cell migration and enhancing cell sensing to its environment. Agrin expression was also found to be linked to actomyosin and F-actin expression, in which siRNA depleted actin expression and sAgrin rescued it. These findings are not surprising based off various previous research finding that ECM is needed for migration or stiffness sensing, which has already been established well for

fibronectin and also found for Agrin *in vitro* (Kawahara, Plos One, 2014). Interestingly, the authors found that culturing the keratinocytes on “cross-bow” shaped geometric constraints further showed how sAgrin supplement could reverse siRNA inhibition, shown specifically by an increase in F-actin and pMLC recruitment.

Many thanks for this comment. Though ECM is needed for stiffness sensing, identifying the key components of the ECM that couple bulk rigidity cues with cellular migration is essential. Kawahara, et.al.¹ and our studies identified that Agrin is required for cancer cell migration via transwell migration assays in 2D *in vitro* assays². However, in this manuscript, we focused on the novel role of Agrin in controlling keratinocytes' mechanoperception and its influence on migration under a mechanically stressed wound environment, both *in vitro* and *in vivo*. Furthermore, its physiological relevance was addressed in skin wound healing. During injury, cells are exposed towards absurd mechanical force, bulk rigidity from the local environment and geometrical tension. Here, we have performed an extensive array of experiments recapitulating each of these mechanical parameters in 2D and 3D-stiffness-dependent contexts to establish that Agrin is a key ECM player regulating keratinocytes' migration under scenarios of bulk ECM rigidity, absurd force and geometrical tension by engaging F-actin and myosin tension (**Figures 2-4**). To the best of our knowledge, this is the first study that demonstrates the mechanistic contributions of Agrin on cell migration under a variety of mechanical stress *in vitro* and in various wound healing mouse models. Another key highlight that differentiates our study from others is the specificity of sAgrin in rescuing the mechanoperception of injured keratinocytes both *in vitro* and *in vivo*. Despite the fact that many proteoglycans and ECM components regulate cell migration in different contexts, the mechanistic insights revealed for Agrin on keratinocytes' mechanoperception have not been identified thus far for any other proteoglycans. Coupled to the notion that sAgrin could be clinically exploited as a promising bio-additive facilitating tissue repair, we strongly believe that our mechanism will be of immense significance to the regenerative medicine and mechanotransduction fields. Therefore, we sincerely hope the reviewer will agree that these new results defining the role, mechanism, function and translational relevance of Agrin in skin wound healing represent scientific insights of sufficient novelty and significance.

3) Finally, they use bulk-RNA sequencing to identify transcriptomic changes in control keratinocytes vs siRNA inhibited ones. They identified MMP12 as gene/protein most closely tied to Agrin expression and repeated previously mentioned experiments to show that MMP12 knockout impaired healing that not even sAgrin treatment could rescue. Finally, the authors treated mouse wounds with sAgrin to show that it speeds up wound closure rates. The authors thoroughly interrogated the effects of Agrin, sAgrin treatment, and MMP12 in their experiments, and their conclusions are sound. However, it is important to note that while these experiments were performed thoroughly and of high quality, their novelty might be limited and they also lacked much physiological relevance, which could limit their impact.

We appreciate that this reviewer considers our data is of high-quality. We have undertaken the following new experimental approaches presented in this revised version to significantly improve the physiological relevance of our findings.

- a) The 2D scratch-wound assays have now been complemented with 3D-stiffness dependent keratinocyte-fibroblasts co-culture wound responsive migration assays.

These 3D assays simulate the migration of injured keratinocytes over dermal fibroblasts under the influence of either a compliant or stiff matrix. Interestingly, Agrin facilitates keratinocytes migration in response to substrate rigidities in these 3D-assays (**Figures 3e-f**).

- b) In addition to the previously shown non-splinted wound healing mouse models, splinted wound healing models have been employed to test the function of sAgrin on the wound microenvironment *in vivo* (**Figure 7**). The use of splinted wound healing models relies on the capacity of sAgrin to facilitate healing by promoting keratinocyte re-epithelization.
- c) Implementation of splinted wound healing models also showed that Agrin depletion attenuated keratinocyte re-epithelization *in vivo* (**Figures 2c-e**).
- d) Extensive *in vitro* and *in vivo* experiments supported that Agrin requires functional MMP12 to restore the mechanical environment of a wound healing niche. This has been recapitulated in 3D-stiffness based co-culture migration assays *in vitro* (**Figure 5f**) and murine skin *in vivo* (**Figure 6**). All of these aspects represent novel findings relating Agrin and MMP12 in skin wound healing by preserving the mechanical architecture of the wound-healing microenvironment. Thus, the Agrin-MMP12 cascade is of originality.

Overall, these lines of investigations further uncover the physiological impact of Agrin in priming the migratory potential of keratinocytes as reflected during re-epithelization under a wound-stressed environment simulated by 3D *in vitro* assays and in splinted models *in vivo*. As mentioned in our response to point 2 above, we sincerely hope the reviewer will find our revised manuscript of sufficient novelty and significance.

4) In terms of physiological relevance, the bulk RNA analysis was performed on cells were cultured on extremely stiff plastic environments, which is not physiologic. Many of the main conclusions were made from scratch assays, which by definition have to be performed on stiff plastic. It would be better if the authors could conduct these *in vitro* wound healing studies in a more physiologically relevant, 3D environment, in which the matrix stiffness more closely matched that of native tissue.

We are thankful to this reviewer for this suggestion. As suggested, we have devised an *in vitro* 3D substrate stiffness dependent migration assay to recapitulate the role of Agrin in mediating keratinocytes' migration under the simultaneous influence of dermal co-cultures and bulk substrate rigidity as experienced by native skin tissues. In this strategy, we first embedded primary mouse dermal fibroblasts (DFs) within a compliant collagen matrix (0.8KPa) which was subsequently overlaid with mouse keratinocytes (KRTs) (**Fig. 3e**). A wound was created at the center and the resulting area was replenished by soft matrix immediately allowing the keratinocytes to migrate through a compliant environment. As shown in **Fig. 3e**, supplementing sAgrin within the soft matrix strongly enhanced keratinocyte migration by day 5 post-injury.

On scenarios recreated on stiff matrix corresponding to 30KPa, Agrin depletion in both KRTs and DFs abolished the migratory rates (**Fig. 3f**, second panel). Interestingly, supplementing sAgrin in this condition strongly rescued keratinocyte migration (**Fig. 3f**, third panel). Of note, Agrin expressed by the DFs was not sufficient to rescue the migration of Agrin depleted keratinocytes over the stiff substrates (**Fig. 3f**, fourth panel). Furthermore, Agrin depleted DFs

had minimal effects on keratinocyte migration over stiff substrates (**Fig. 3f**, fifth panel). Together, these data indicate that increased Agrin expression within keratinocytes majorly responds to ECM rigidity sensing to drive migration. Through this approach, we addressed that Agrin controls the keratinocytes' migration under the influence of appropriate substrate stiffness and fibroblasts.

Our additional set of experiments using this approach further confirmed that Agrin requires MMP12 in keratinocytes and fibroblasts for mediating keratinocytes' migration in response to injury. As shown in revised **fig. 5f**, sAgrin incorporated soft matrix stimulated mouse keratinocyte migration over dermal fibroblasts post-injury (**Fig. 5f**). In contrast, depleting MMP12 in mouse keratinocytes and underlying dermal fibroblasts abrogated the keratinocyte migration which was not rescued by sAgrin (**Fig. 5f**). These assays were performed with mouse primary keratinocytes and fibroblasts to mimic the response of wounded skin more closely *in vitro*.

Cumulatively, the novel approach presented here confirm that Agrin regulates the keratinocytes' migration by engaging MMP12 under the influence of fibroblasts and bulk ECM rigidity with the role of Agrin and MMP12 executed primarily from keratinocytes.

5) Furthermore, authors identified keratinocytes as the primary drives of wound closure, but they did not use an animal model that relies on re-epithelialization. It is well known that mice have a panniculus carnosus muscle under their skin, leading them to heal mostly by contracting their wounds. As such, most mice heal their wounds by a quick "purse string" like action, unlike that seen in humans or other large animals. It would be better for the authors to use methods such a splinting in order to promote a more human-like wound healing environment that is driven by re-epithelialization by the keratinocytes (Galiano, Wound Repair and Regeneration 2004). This would bring more confidence that their findings are actually primarily driven by keratinocytes and also significantly increase the translational impact of their findings.

Many thanks for this insightful suggestion. As suggested, we have conducted several of our key experiments using the splinted wound healing model devised by Galiano et. al. ³. The new results presented below strongly enhance the impact of our findings:

- a) First, we tested whether Agrin induced during murine skin injury controls keratinocytes' re-epithelization and subsequent healing. The 4 mm wounds were surrounded by a 10 mm splint adhered to the skin, thereby representing a 'closed' splinted condition for wound healing. The ointments containing the respective siRNAs were applied and subsequently the wound region were covered by Tegaderm (**Fig. 2c**). As such, the usage of splints minimized the 'purse-string' mediated wound contraction and majorly facilitated wound closure by re-epithelization. In addition, covered splinted wound dressings reduced wound healing rates at day 7 when compared to those in non-splinted conditions (**Fig. 2c**). Using this *in vivo* model as an index for measuring keratinocytes' migration, we showed that Agrin depletion delayed skin wound healing by attenuating keratinocyte re-epithelization and K17 occupancy (**Figs. 2d-e**). In addition to impaired re-epithelization, Agrin depletion severely dampened the deposition of mature and intermediate collagen fibers in the wound beds, indicating

that compromised ECM replenishment in Agrin depleted skin resulted in delayed healing response (**Fig. 2f**).

- b) We determined the wound healing capacity of sAgrin using this splinted model (**Fig. 7**). In comparison to BSA or collagen (used as controls), sAgrin significantly accelerated the healing rates of injured murine skin. This was achieved by improved re-epithelization, proliferation of wound margins and restoring the MMP12-pMLC engaged mechanical tension of keratinocytes in the wounded microenvironment. In addition to restoring ECM deposition in the wound beds, we also showed that sAgrin triggered a pro-angiogenic environment that facilitated healing rates under splinted conditions. These new data are incorporated as **Figs. 7b-h**, respectively.
- c) Furthermore, the role of MMP12 as a downstream mediator of Agrin's wound healing mechanics was tested *in vivo* via the splinted wound healing model. The new results presented in **Figs. 6a-e** shows that Agrin fails to heal wounds in MMP12 deficient mouse skin tissues. In addition, MMP12 depletion renders a mechanically incompetent environment that is deprived of ECM replenishment (as suggested by decreased collagen deposition- **Fig 6d**), acto-myosin mechanotension (**Fig. 6c**) and reduced angiogenesis (**Fig. 6e**) that is no longer responsive to sAgrin.

Therefore, Agrin promotes keratinocytes re-epithelization and accelerates wound repair by engaging MMP12 as shown in the splinted wound healing mouse models. In addition, the relevance of sAgrin as a regenerative skin bio-additive was also showcased in splinted wound healing models that closely relies on efficient re-epithelization.

6) In terms of novelty and impact of the findings, previous research (Hattori, The American Journal of Pathology, 2009) has already identified both MMP9 and MMP13 as critical drivers of wound closure in keratinocytes using a mouse wounding model without splinting. The novelty of Agrin as a potential mechanotransduction regulator is also brought into question from a search from the literature. The authors clearly have experience studying Agrin and its key role in mechanotransduction and force sensing in a variety of other organ systems such as liver (Chakraborty, Cell Reports, 2017, Chakraborty, Nat Comm, 2015). Agrin has also been found by others to promote myocardial regeneration (Bassat Nature 2017). While the role of Agrin and its link to MMP12 in keratinocytes related to dermal healing has not been specifically investigated, other have recently shown that Agrin promotes corneal wound healing (Hou, Investigative Ophthalmology & Visual Science, 2020).

We thank the reviewer for this comment. As described previously, the novelty of Agrin in priming the wound healing mechanics have not been elucidated thus far in skin or any other tissues *per se*. The evidence that Agrin may serve a mechanotransducer of absurd extrinsic force, geometrical tension and bulk substrate rigidities that overhauls the keratinocyte mechanics during mechanical injury is of prime interest in this report (**Figs. 4e-i, S6**).

More importantly, the mechanistic insights that Agrin acts via engaging MMP12 as downstream effector is of originality. As the reviewer mentioned, MMP9 and MMP13 have been reported to control skin keratinocytes' migration and wound healing along-with roles in other tissue healing as well. In addition, we would like to point out that several MMPs have profound role(s) in regulating proteoglycan levels during healing⁴. However apart from serving as docking sites for MMPs, key ECM proteoglycans driving MMP expression during stages of

wound healing is not clearly understood, particularly in the context of priming wound healing mechanics. Several salient features distinguish the role of Agrin-MMP12 axis in the skin microenvironment which advanced our understanding of the complex wound environment. In our study, we provide the first evidence that early induction of Agrin specifically activates MMP12 for restoring the mechanical environment required for wound healing and that MMP12 is an important mediator of Agrin for skin re-epithelization and accelerated healing. As shown by the Agrin-MMP12 pathway, preserving the mechanoarchitecture of the wound microenvironment has not been previously associated with any MMPs or proteoglycan axis in tissue repair. Specifically, our *in vitro* data reveals that Agrin-MMP12 coordinates a fluidic collective migration that is essential for wound healing process *in vivo* (**Figs. 5j, S9l-m**, respectively). In addition to the discovery of a novel Agrin-MMP12 axis, these shreds of evidence also uncover a previously uncharacterised role of MMP12 in controlling keratinocyte re-epithelization, mechanical tension, ECM deposition and angiogenesis in the skin wound bed as a downstream mediator of Agrin (new data in **Fig. 6**).

In addition, we present the first evidence that recombinant versions of Agrin (sAgrin) may be incorporated as bio-additive for skin regenerative scaffolds. As such, our data reveal that sAgrin provides optimal mechanobiological and angiogenic signals to accelerate wound healing (**Fig. 7**). Furthermore, sAgrin was shown to induce a highly selective set of pro-inflammatory cytokines that lay foundation for an efficient healing program (**Fig. S12**). In addition to cutaneous wound healing, Agrin has been shown to be important for regeneration of infarcted myocardial muscles and cornea with precise mechanisms remaining less clear. Whether Agrin overhauls the mechanical parameters has never been investigated in the context of cardiac or corneal wound healing. Hence, the mechanisms presented here may offer valuable insights to cardiac and corneal healing for future investigations. Therefore, we sincerely hope the reviewer share our understanding that our study provides a strong mechanistic basis for Agrin-induced regeneration and clinical relevance for using sAgrin as a bio-additive.

Some minor concerns:

a) The authors claim Agrin increases in mRNA content, but the figure in 1B does not always show that trend. Additionally, no statistics seem to have been calculated.

Thank you for pointing this out. We have redone the mRNA expression profile(s) of Agrin (AGRN), Glypican-1 (GPC1) and Perlecan (HSPG2). The new results presented as **Fig. 1b** show that the levels of Agrin are consistently increased within 3-24h post-injury in keratinocytes (HaCaT and HEK); while the respective levels in fibroblasts (BJ) increased only at later stages (i.e. 24h post-wounding). In contrast, GPC1 levels are increased only in HaCaTs but not in HEK or BJ cells post-injury. HSPG2 did not show any significant increase in any of the analysed cell lines within the observed 24h. The statistical analysis is now computed and shown in revised panel **Fig. 1b**.

b) It seems like the mouse skin implants did not show any significant difference (figure 2g). I would like the authors to comment on why that might be.

Many thanks for this comment. Our revised panels shown as **Fig. 3i** show significant reduction in the keratinocyte outgrowth of Agrin depleted skin implants cultured on stiff substrates ($p=0.019$). In addition, the revised figure also depicts significant rescue of keratinocyte outgrowth by sAgrin ($p=0.04$) in this condition. We have included the figure panel for the reviewer's easy reference.

c) Figure panel organization was sometimes hard to follow, especially since the treatment groups were arranged by column and other times by rows.

Thanks for the input. We have reorganized the figure panels in revised **Fig. 7** to maintain consistency.

d) The authors chose MMP12 since it had the highest significance, but MMP1 and MMP10 also seemed to have interestingly high fold changes that would also be interesting to evaluate. The authors should mention if they tested this or why they did not interrogate that expression.

Once again, we thank this reviewer for this constructive comment. Agrin depletion significantly reduced the levels of MMP1, 10 and 12 in our transcriptome-wide RNASeq analysis. We have now included additional data that rule out any significant effects of MMP1 and MMP10 as downstream mediators of Agrin in governing the mechanical tension post-wound injury. First, we evaluated whether MMP 1, 10 and 12 independently affected keratinocyte migration following wound-scratch. Accordingly, knocking down MMP 1 and 10 reduced keratinocytes' migration that was not restored by sAgrin treatment, suggesting they are involved in Agrin-mediated functions (**Fig. S8a**). In comparison to MMP1 and 10, MMP12 knockdown showed the maximal inhibition of migration velocities post-injury (**Fig. S8b**). Unlike MMP12, we did not observe that the mRNA levels of MMP1 and MMP10 were consistently upregulated following wound injury in HaCaT cells (**Fig. S8c and Fig. S9c**). Most importantly, MMP1 or MMP10 depletion had no effects on the acto-myosin tension at the wound edges (**Fig. S8d**). When cultured on large crossbow patterns, MMP1 or MMP10 depleted cells did not impact the tension signatures of F-actin stress fiber network and pMLC enriched transverse arcs (**Fig. S8e**). Despite being regulated by Agrin, these observations suggests that MMP1 or 10 are not major downstream mediators in regulating the wound healing mechanics. While MMP1 and MMP10 may have role(s) in controlling keratinocyte migration post-injury, our data suggests

that Agrin preferentially engages MMP12 for mediating mechanical tension in the wound microenvironment.

We have also mentioned the following lines in the results section on pages 14-15: 'Despite being regulated by Agrin, these observations suggests that MMP1 or 10 are not major downstream mediators in restoring the wound healing mechanics. Despite the fact that several MMPs are involved in cancer cell migration, matrix crosslinking, and wound healing, a concrete role of MMP12 has not been elucidated in mechanotransduction involved during cutaneous wound repair. As such, MMP12 emerged as the most significant effector for mediating Agrin's mechanotension in wound repair.' Therefore, our study revealed a new role and underlying mechanism for MMP12 to mediate the role of Agrin in wound healing.

Reviewer #2 (Remarks to the Author):

The manuscript entitled, "Agrin-Matrix Metalloproteinase-12 axis confers a mechanically competent microenvironment in skin wound healing" is required major revision before considering this manuscript for publication.

Many thanks to the reviewer for the constructive comments which have guided us to improve our study significantly.

Comments:

1. What about the purity of Agrin.

Thanks for this input. We have now added additional data to show the purity of recombinant Agrin (sAgrin) in **Fig. S11a**. The coding region for the C-terminal fragment containing 8 amino acid 'Z8' insert and LG3 domain was cloned into a pET28 vector containing His-tag and expressed in Escherichia coli (strain BL21-DE3). The bacteria were transformed with the plasmid. A single colony was inoculated in TB medium containing antibiotics and induced with IPTG for 16h at 15°C. The cells were harvested by centrifugation and cell pellets were lysed in lysis buffer followed by sonication. The resultant supernatant containing Agrin protein fragment was purified using gel filtration chromatography via Superdex75 columns in imidazole buffer, and sterilized by passing through a 22µm low-protein binding filter and was dissolved in PBS solution (**mentioned in the methods section**). The gel-filtration profile of sAgrin is shown in **Fig. S11a** and the protein yield was ~18mg/l. Moreover, we have detected the purified sAgrin in a Coomassie-brilliant blue stained gel (**Fig. S11a**) and by Western blot analysis (**Fig. S11a**). As evidenced by Coomassie-brilliant blue stained gel, sAgrin is of sufficient purity.

2. How was the stiffness analyzed? By Rheology? This details should be given.

We thank the reviewer for this suggestion. We measured cell and substrate stiffness by Atomic Force Microscopy (AFM). The stiffness of substrates used for traction force microscopy was also measured by AFM and is described in the methods section. The stiffness of silicone polyhydrogels were measured by the manufacturer (Advanced Biomatrix, Inc.) as per their protocol ⁵. All of this information is presented in detail in the respective methods section 'Substrate stiffness manipulations and micropatterning', 'Traction Force Microscopy' and 'Cell stiffness measurements by Atomic Force Microscopy (AFM)' on pages 30 and 33-35, respectively.

What is the stiffness is needed for wound regeneration.

Many thanks for this insightful comment. We have performed AFM measurement of stiffness on migrating mouse primary keratinocytes, the key drivers of re-epithelization post-wound injury. The stiffness of mouse primary keratinocytes that actively migrate during wound closure at 4h post-injury was found to be ~2.1KPa. Knocking down Agrin in these keratinocytes hampered their migration rates and reduced the intrinsic cell stiffness to ~0.9KPa. Consistently, supplementing sAgrin to Agrin depleted cells substantially increased cell

stiffness to ~1.5KPa. These new results are presented as **Fig. 3j**. The respective atomic force stiffness maps of cells are shown in **Fig. S5b**. As such, Agrin is important for maintaining the stiffness that support keratinocyte migration.

Furthermore, we also evaluated whether MMP12 depletion affected the cell stiffness during wound regeneration. The results shown in **Fig. S9I** revealed that loss of MMP12 in these migrating keratinocytes hampered their cell stiffness, which was not restored by sAgrin treatment. Cumulatively, these new data illustrate that Agrin bestows significant cell stiffness by engaging MMP12 during wound regeneration.

3. Size of the wound is not mentioned. It is very important. Any approved protocol is used for the wound creation model.

Thank you for this input. We have mentioned the size of the wound as 4mm in the methods section 'Punch-biopsy wound healing models' and in the results section on pages 6-7. Moreover, we have included detailed explanation of both 'non-splinted' and 'splinted' versions of wound healing models in the results and methods sections. The wound-healing IACUC protocol (internal protocol approval #171237) is also explained in the methods section as follows: 'For both models, the animals were caged individually throughout the observed time-points of healing at the institutional animal facility. All wound healing animal experiments performed were done in accordance with experimental protocols reviewed by the Biological Resource Center (BRC), Agency for Science Technology and Research (A*STAR) under strict compliance to the Institutional Animal Care and Use Committee (IACUC) guidelines for ethical use of animal models in biological research'.

4. What about angiogenic nature of Agrin. It should be discussed.

We thank this reviewer for this constructive suggestion. As shown in our revised version, we have undertaken several experimental approaches to support that Agrin promotes a pro-angiogenic niche in the wound beds, thereby facilitating accelerated wound repair.

Firstly, new results presented as **Fig. 7h** demonstrate that treatment with sAgrin significantly improved the number and sizes of CD-31+ve blood vessels in the murine skin wound bed in both non-splinted and splinted mouse models.

Secondly, reduced angiogenesis (fewer blood vessels) was also observed within the wound beds of MMP12 depleted mouse skin (**Fig. 6e**). As such, the treatment of sAgrin failed to rescue CD-31 positive blood vessels when MMP12 was suppressed in the mouse skin (**Fig. 6e**). Together, these observations reveal that sAgrin requires MMP12 to generate a mechanically competent pro-angiogenic wound healing niche *in vivo*. (**Fig. 6e**).

Thirdly, our *in vivo* data suggests that Agrin may boost the interaction of dermal fibroblasts and endothelial cells, thereby enhancing angiogenic activity within the wound bed. To test this *in vitro*, we depleted Agrin in human dermal fibroblasts (GFP-tagged BJ) and human dermal microvascular endothelial cells (cell-tracker blue labelled HDMEC) and subjected them to a 3D co-culture sprouting angiogenesis assay (**Fig. S13a**). Notably, the control spheroid co-cultures generated robust sprouting which was significantly diminished by suppressing Agrin in endothelial cells and fibroblasts individually (**Fig. S13b**, second and third panels,

respectively). Consistently, depleting Agrin simultaneously in the fibroblasts and endothelial cells resulted in stronger inhibition of sprouting angiogenesis (**Fig. S13b**). Together, these results indicate that Agrin supports a pro-angiogenic environment in the wound bed by fostering interactions between the dermal fibroblasts and endothelial cells. These are interpreted in the results section on pages 20-21.

5. Collagen quantification is not shown in histological analysis.

We are grateful to the reviewer for pointing this out. As suggested, we have performed picosirius red staining and quantifications of mature, mixed and immature collagen fiber depositions in the wound beds. In the first instance, we report that Agrin depletion in the murine skin severely dampened the deposition of mature and intermediate collagen fibers in the wound beds (**Fig. 2f**). Second, sAgrin treatment induced mature and mixed collagen fiber deposition *in vivo*. The effect was suppressed upon MMP12 depletion at the wound beds (**Fig. 6d**). Thus, MMP12 depletion renders a mechanically incompetent environment that is deprived of ECM replenishment and is no longer responsive to sAgrin. Third, as suggested by the reviewer (see point #8), we have compared the relative collagen deposition rates promoted by sAgrin in comparison to BSA or rat-tail collagen as controls. The new results presented as **Fig. 7g** show that sAgrin treated wound beds exhibited higher deposition of mature and mixed collagen fibers when compared to BSA and Collagen treated groups (**Figs. 7g**). Similar enhancement of collagen deposition by sAgrin was observed within the wound beds at day 7 in non-splinted conditions (**Fig. S11i**). Additionally, these shreds of data also suggest that sAgrin promotes optimal collagen deposition in the wound bed to sustain the degree of stiffness within the wound healing niche.

6. How this Agrin contribution in the infectious wounds or diabetic wounds. It should be discussed in the manuscript.

We appreciate this suggestion from the reviewer. Accordingly, we have mentioned the following lines in discussion section on page 22: 'Fourthly, appreciating the importance of a mechanically competent Agrin enriched wound healing environment, our results additionally reveal that a recombinant soluble Agrin fragment may further accelerate the wound healing process. Hence, appropriate utilization of sAgrin as a bio-additive wound healing material may offer potential clinical value for wounds that are naturally reluctant to heal, as presented by diabetic or burn-injury patients. Furthermore, our study would pave the way to investigate whether Agrin levels are decreased in the skin tissues of diabetic patients and those infected with bacterial infections, presenting a basis for chronic delayed healing of wounds in these patients. Subsequent studies involving sAgrin as a wound healing bio-additive in diabetic mouse wound healing models will also reveal the potential in future translational research among diabetic patients.'

7. What about the quantifications of inflammatory cytokines in the wounds after the treatment.

Many thanks for this input. We have performed additional experiments to compile the profile of inflammatory cytokines induced by sAgrin in the murine skin during the early phases of

wound healing. We profiled the mRNA expression of several cytokines and chemokines known to be induced within 48 hours after injury to the mouse skin ⁶. Among them, TGF- β 1, VEGF-A, MCP-1, MIP-1 β and MIP-2 were induced in the mouse skin wounds within 48h post-injury that received sAgrin treatment in comparison to BSA or Collagen treated groups (**Fig. S12a**). Indeed, augmented protein expression(s) of TGF- β 1, VEGF-A and MCP-1 were confirmed in sAgrin treated groups, when compared to BSA and Collagen treated mouse skin tissues (**Fig. S12b**). Therefore, timely engagement of a selective set of pro-inflammatory response by sAgrin is consistent with an accelerated wound healing program. The ability of sAgrin to induce TGF- β 1, VEGF-A and MCP-1 as part of pro-inflammatory program is also novel when coupled to the consequence of improved healing parameters.

8. I also suggest that collagen gel can be used as one control in the animal studies.

Thank you for this excellent suggestion. We have performed additional experiments comparing the efficacy of sAgrin with rat-tail collagen-type I and BSA in accelerating wound repair employing a splinted wound healing mouse model. In this study, sAgrin, collagen or BSA were incorporated as bio-additives using Pluronic-F127 based hydrogel ointment. The new results are presented in **Figs. 7b-h**. The incorporation of 200ug sAgrin within Pluronic F127 significantly accelerated wound healing when compared to similar concentrations of collagen or BSA during the observed 12 days post-injury (**Fig. 7b**). In splinted conditions at day 10 post-wounding, sAgrin treated animals predominantly represented healed skin with the emergence of hair follicles in the vicinity of the wound area in comparison to BSA or Collagen treated groups (**Fig. 7c**). As wound healing required a longer time-frame under splinted conditions, we accounted a similar increment of K17 occupancy, MMP12 expression and pMLC activity at day 7 post-injury in sAgrin treated animals in comparison to BSA or Collagen counterparts (**Figs. 7d-f**). Since ECM replenishment within the wound bed guides the overall healing process, sAgrin treated wound beds had higher deposition of mature and mixed collagen fibers when compared to BSA and Collagen treated groups (**Figs. 7g and S11i**). Additionally, these shreds of data also suggest that sAgrin promotes optimal collagen deposition in the wound bed to sustain the degree of stiffness within the wound healing niche (**Figs. 7g and S11i**).

Moreover, our results also report that sAgrin treatment induces a selective set of inflammatory cytokines in comparison to the Collagen treated group as described in our response to point 7.

Reviewer #3 (Remarks to the Author):

In this work “Agrin-Matrix Metalloproteinase-12 axis confers a mechanically competent microenvironment in skin wound healing”, Sayan Chakraborty and colleagues investigate if and how Agrin promotes cells migration after skin wounding to stimulate process of healing. Their use of animals in vivo, skin ex vivo, and in vitro assays and human explants/cells approaches is a strength of their work. Moreover, the extension of methods including those focused on Traction force microscopy and Force transmission via ligand conjugated magnetic beads sheds new mechanistic insights into how Agrin enhances the mechanoperception of keratinocytes thereby promoting cell migration participating the skin wound healing process. Additionally, the data showed that Agrin has a promising potential as a bio-additive material for accelerated wound healing. Although the coverage of this manuscript is extensive, the results, particular those related to Agrin-Mmp-12 pathway and the mechanisms are rather descriptive. Usage of animal models as Mmp-12KO mice since Agrin KO is lethal would be of value. Additionally primary culture of mouse cells as a major tool followed by human cell lines would strengthen the data.

Many thanks to this reviewer for appreciating our work and providing a constructive feedback. We have now addressed the previous comments by new set of experiments that significantly improved our study.

1. Figure 1a shows the changes in gene-expression in non-wounded vs post-wounded mouse skin (single-cells transcriptomics analysis). The changes in AGRN expression was not profound as for GPC1 or 2. However in in vitro migration model it is AGRN which showed the most substantial changes, please explain/comment on it.

Thank you for pointing this out. We agree with the reviewer that the increase in the levels of GPC1 (Glypican1) was higher than that observed for Agrin in the analysed wounded mouse skin cells via single-cell transcriptomic data. While it is clear that GPC1 levels are strongly induced in the wounded mouse keratinocytes, cultured human keratinocytes (except HaCaT) and fibroblasts did not show a consistent increase in GPC1 levels (**Fig. 1b**). In addition, wound injury to HaCaT cells strongly induced GPC1 mRNA levels similar to that observed in the wounded mouse keratinocytes analysed in Fig. 1a. Therefore, we have further performed scratch-wound migration assays comparing the degree of inhibition caused by GPC1 or AGRN knockdown individually in HaCaT cells. The new results shown in **Fig. S1a-c** reveal that GPC1 knockdown caused a greater inhibition of HaCaT cell migration and their respective velocities compared to that of AGRN knockdown. These data are consistent with the level of induction observed upon injury in HaCaT cells and wounded mouse keratinocytes shown in Fig. 1a-b. We have interpreted these results and the reason for focussing on Agrin on page 5: ‘Consistent with the data analyzed in Fig 1a, we observed that depleting GPC1 displayed greater inhibition of HaCaT cell migration velocities post-wounding when compared to that of Agrin knockdown (Fig. S1a-c). Despite the fact that Agrin expression was induced robustly in both keratinocytes and fibroblasts upon injury (Fig. 1b), GPC1 may also serve as an important ECM proteoglycan promoting skin wound healing. These are further substantiated by defined prior roles of Glypicans and Perlecans in skin wound healing ^{7,8}.’

2. Which cells in the skin are the major source of agrin?

It would be important to show how the content of agrin changes in relation to epidermis, dermis (papillary, reticular and dWAT-component) and how the agrin location/abundance is affected by wound healing process?

Many thanks for this suggestion. Results from new experiments confirmed Agrin expression in keratinocytes and dermal fibroblasts. Firstly, we observed that Agrin is highly expressed in keratinocytes in comparison to fibroblasts (of both human and mouse origin) *in vitro*. The new results are presented as **Fig. 1d**. Even in non-wounded states, both human and mouse keratinocytes expressed higher Agrin levels when compared to their respective dermal fibroblasts, implying that keratinocytes sourced Agrin may play more critical roles in skin wound repair (**Fig. 1d**).

Secondly, as suggested by the reviewer, we have analysed the changes of Agrin expression via immunohistochemistry in intact or wounded mouse skin. This further revealed a significant increase of Agrin expression within the keratinocyte layers of the epidermis in comparison to the dermis at day 2 that maximized by day 4 post-wound injury (**Fig. 1f**). While increased Agrin expression was also observed within the injured dermis layers between days 2-4, no significant change was detected in the hypodermis and dermal-white adipose tissues (D-WAT) at any stage post-injury (**Fig. 1f**). Together, these results revealed that Agrin expression is significantly triggered within the epidermal and dermal layers of skin upon mechanical injury.

3. It would be of value in the cell migration (scratch wound model) use primary mouse skin cells: keratinocytes and DFs, since in vivo data are based on mouse model. Moreover, there are the differences in AGRN protein abundance between HaCaT cells (established cell line) and HEK (primary immortalized line) therefore applying primary keratinocyte/DFs cultures from mouse skin would be of value.

Thank you for the excellent suggestion. We agree that using primary mouse cultures in addition to human cell lines would increase the impact of our finding. Accordingly, we have performed additional wound healing experiments using primary mouse keratinocytes (KRT) and dermal fibroblast (DF) cultures obtained from C57BL mouse strain. Firstly, we showed that primary mouse keratinocytes have higher Agrin expression when compared to dermal fibroblasts (**Fig. 1d**). Wound injury to mouse primary keratinocytes also showed a consistent increase in Agrin and MMP12 levels (**Figs. 1c and S9c**). Next, we depleted Agrin by using mouse specific siRNAs in these keratinocytes and fibroblasts to see the impact on 2D-scratch-wound assays. The results presented as **Fig. S3b-c** showed that depletion of Agrin hampered the migratory abilities of these primary mouse cultures, which were significantly restored by sAgrin supplementation. Moreover, we noticed a significant decrease in the migratory velocities of Agrin depleted mouse KRTs and DFs that was again rescued by sAgrin (**Fig.S3d**).

In addition, we devised an *in vitro* 3D substrate stiffness dependent migration assay to recapitulate the role of Agrin in mediating keratinocytes' migration under the simultaneous influence of underlying dermal cultures and bulk substrate rigidity as experienced by native skin tissues. In this strategy, we first embedded primary mouse dermal fibroblasts (DFs) within

a compliant collagen matrix (0.8KPa) which was subsequently overlaid with mouse keratinocytes (KRTs) (**Fig. 3e**). A wound was created at the center and the resulting area was replenished by soft matrix immediately allowing the keratinocytes to migrate through a compliant environment. As shown in **Fig. 3e**, supplementing sAgrin within the soft matrix strongly enhanced keratinocyte migration by day 5 post-injury.

On stiff matrix, Agrin depletion in both KRTs and DFs suppressed the migratory rates (**Fig. 3f**, second panel). Interestingly, supplementing sAgrin in this condition strongly rescued keratinocyte migration (**Fig. 3f**, third panel). Of note, Agrin expressed by the DFs was not sufficient to rescue the migration of Agrin depleted keratinocytes over the stiff substrates (**Fig. 3f**, fourth panel). Furthermore, Agrin depletion in DFs had minimal effects on keratinocyte migration over stiff substrates (**Fig. 3f**, fifth panel). Together, these data indicate that increased Agrin expression within keratinocytes majorly responds to ECM rigidity sensing to drive migration.

Additional experiments using this method further confirmed that Agrin requires MMP12 in keratinocytes and fibroblasts for mediating keratinocytes' migration in response to injury. As shown in revised **fig. 5f**, sAgrin incorporated soft matrix stimulated mouse keratinocyte migration over dermal fibroblasts post-injury as shown in 3D stiffness-dependent migration assays (**Fig. 5f**). In contrast, depleting MMP12 in mouse keratinocytes and underlying dermal fibroblasts clearly inhibited the keratinocyte migration which was not rescued by sAgrin (**Fig. 5f**).

Cumulatively, new data presented here confirm that Agrin regulates the keratinocytes' migration by engaging MMP12 under the influence of fibroblasts and bulk ECM rigidity.

4. How the authors separate cells migration from cells proliferation during in vitro assay. Re-epithelialization process in vivo is composed by migrating epithelial tongue that is pushed from the back by proliferating keratinocytes. However, in in vitro model the cells migrate and proliferate simultaneously.

This is an excellent insight from the reviewer. This was addressed by detailed proliferation based *in vitro* and *in vivo* experiments that indicate a role of Agrin in supporting keratinocytes' proliferation during their migratory phase. During early *in vitro* migration within 4h post-wound injury simulated by a 2D-scratch wound assay, Agrin depletion did not affect the proliferative rates of the leader cells (at the wound edge) as measured by 5-bromo-2'-deoxyuridine (BrdU) incorporation assays (**Fig. S4a**). However, the proliferation of follower cells away from the wound margin was significantly reduced by Agrin knockdown, an effect which was rescued by sAgrin supplementation (**Fig. S4a**). These experiments were consistently replicated in both human (HaCaT) and primary mouse (KRT) keratinocytes. Hence, our *in vitro* experiments reveal that Agrin differentially affects the proliferative states of leader and follower cells during wound-injury.

In vivo, an efficient wound re-epithelialization is achieved when highly proliferative keratinocytes (followers) at the wound edge push the migrating cells (leaders) over the wound (**Fig. S4b**). As shown in the control siRNA treated wound sections, proliferative keratinocytes at the base of the epidermis (yellow arrows) are actively pushed during re-epithelialization by the hyper-proliferative wound edge regions (red arrows) at day 7 (**Fig. S4b**). Coupled to impaired

proliferation at the base of epidermis, Agrin depletion severely hampered the keratinocyte proliferation in the dermal layers at the wound edge and beds (**Figs. S4b-c**). Therefore, these shreds of evidence suggest that Agrin enriched in the wound environment supports keratinocyte proliferation that primes an efficient collective migration phase and wound closure.

5. Immunofluorescent detection of agrin in mouse skin should be also confirmed by immunohistochemical detection in non-wounded and post-wounded skin. IHC detection more clearly should show the abundance in protein localization in the case of agrin epidermis vs dermis in relation to days of injury.

Many thanks for this suggestion. As explained in our response to comment #2, we have analysed the changes of Agrin expression via immunohistochemistry in intact or wounded mouse skin. This further revealed a significant increase of Agrin expression within the keratinocyte layers of the epidermis in comparison to the dermis at day 2 that maximized by day 4 post-wound injury (**Fig. 1f**). While increased Agrin expression was also observed within the injured dermis layers between days 2-4, no significant change was detected in the hypodermis and dermal-white adipose tissues (D-WAT) at any stage post-injury (**Fig. 1f**). Together, these observations revealed that Agrin expression is significantly triggered within the epidermal and dermal layers of skin upon mechanical injury.

6. Using *in vivo* model of 4mm diameter wound on the back of mice and then applying scrambled or Agrin siRNA in the ointment at the wound site how you protect animals from the liking ointment off the skin?

We appreciate the reviewer's concern here. Accordingly, we have employed an additional *in vivo* procedure to test the effects of Agrin in mediating keratinocyte re-epithelization via employing a 'splinted' wound healing. In this model, the 4 mm wounds at the shoulder skin of mouse were surrounded by a 10 mm transparent splint adhered to the skin (dressing scheme presented in **Fig. 2c**) The ointments containing the respective siRNAs were applied and subsequently the wound region were covered by Tegaderm that prevents the animals from leaking off the dressing (**Fig. 2c**). This model represents a closed mode of healing that relies majorly on keratinocyte mediated epithelization rather than wound contraction. As such, the usage of splints minimized the 'purse-string' mediated wound contraction and majorly facilitated wound closure by re-epithelization. In addition, covered splinted wound dressings reduced wound healing rates at day 7 when compared to those in non-splinted conditions (**Fig. 2c**). Using this *in vivo* model as an index for measuring keratinocytes' migration, Agrin depletion delayed skin wound healing by attenuating keratinocyte re-epithelization and K17 occupancy (**Figs. 2d-f**).

Furthermore, we also tested the efficacy of sAgrin using similar splinted models that protected the animals from leaking off the ointments. These results are presented in **Fig.7b-h**, respectively.

7. There are no discussion concerning agrin presence in keratinocytes vs dermal fibroblasts and the possible DFs effect on the process?

Many thanks for this constructive feedback. We have performed experiments to address the possible effects of mouse DFs on the migrating KRTs. As discussed in point #3, we devised an *in vitro* 3D substrate stiffness dependent migration assay to recapitulate the role of Agrin in mediating keratinocytes' migration under the simultaneous influence of dermal co-cultures and bulk substrate rigidity as experienced by native skin tissues. In this system, we first embedded primary mouse dermal fibroblasts (DFs) within a compliant collagen matrix (0.8KPa) which was subsequently overlaid with mouse keratinocytes (KRTs) (**Fig. 3e**). A wound was created at the center and the resulting area was replenished by soft matrix immediately allowing the keratinocytes to migrate through a compliant environment. As shown in **Fig. 3e**, supplementing sAgrin within the soft matrix strongly enhanced keratinocyte migration by day 5 post-injury. On stiff matrix, Agrin depletion in both KRTs and DFs robustly abolished the migratory rates (**Fig. 3f**, second panel). Consistently, supplementing sAgrin in the stiff matrix strongly rescued keratinocyte migration (**Fig. 3f**, third panel). Of note, Agrin expressed by the DFs was not sufficient to rescue the migration of Agrin depleted keratinocytes over the stiff substrates (**Fig. 3f**, fourth panel). Furthermore, Agrin depleted DFs had minimal effects on keratinocyte migration over stiff substrates (**Fig. 3f**, fifth panel). Together, these data indicate that increased Agrin expression within keratinocytes majorly responds to ECM rigidity sensing which is necessary and sufficient to drive migration.

Furthermore, we have also mentioned this our discussion on page 23 stating 'While Agrin depletion independently inhibited 2D scratch wound healing in keratinocytes and dermal fibroblasts, 3D stiffness-dependent keratinocyte fibroblast co-culture migration assays revealed that dermal fibroblasts sourced Agrin is insufficient to restore the migration of Agrin depleted keratinocytes on stiff substrates. Also, the knockdown of Agrin in the dermal fibroblasts did not impact the migratory rates of overlaid keratinocytes with intact Agrin on stiff substrates. Given that non-wounded and wounded keratinocytes expressed higher levels of Agrin than in dermal fibroblasts, Agrin expressed by keratinocytes acts as a master regulator coordinating migration over wounded regions upon sensing bulk ECM rigidity.'

8. Based on HaCaT cells there are differences in the expression not only for Mmp-12 but also for Mmp-10 and Mmp-1 (that in mice is rather Mmp-13) why the Mmp-12 was chosen? It is not clear.

Many thanks, we have addressed this in point #9 below.

9. To exclude that Mmp-10 or Mmp-1 (Mmp-13) is involved in the Agrin action the same set of experiments as for mmp-12 should be performed.

Similar to reviewer 1, we thank this reviewer for this constructive comment. We carried out similar set of experiments for MMP1 and MMP10 as shown below. Agrin depletion significantly reduced the levels of MMP1, 10 and 12 in our transcriptome-wide RNASeq analysis. We have now included additional evidence that rule out any significant effects of MMP1 and MMP10 as downstream mediators of Agrin in the mechanical tension post-wound injury (**Fig. S8**). First, we evaluated whether MMP 1, 10 and 12 independently affected keratinocyte migration following wound-scratch. Accordingly, knocking down MMP 1 and 10 reduced keratinocytes' migration that was not restored by sAgrin treatment, suggesting they are involved in Agrin-mediated functions (**Fig. S8a**). In comparison to MMP1 and 10, MMP12 knockdown showed the maximal inhibition of migration velocities post-injury (**Fig. S8b**). Unlike MMP12, we did not

observe that the mRNA levels of MMP1 and MMP10 were consistently upregulated following wound injury in HaCaT cells (**Fig. S8c** and **Fig. S9c**). Most importantly, MMP1 or MMP10 depletion had no effects on the acto-myosin tension at the wound edges (**Fig. S8d**). When cultured on large crossbow patterns, MMP1 or MMP10 depleted cells did not impact the tension signatures of F-actin stress fiber network and pMLC enriched transverse arcs (**Fig. S8e**). Despite being regulated by Agrin, these observations suggests that MMP1 or 10 are not major downstream mediators in the wound healing mechanics. While MMP1 and MMP10 may have role(s) in controlling keratinocyte migration post-injury, our data suggests that Agrin preferentially engages MMP12 for mediating mechanical tension in the wound microenvironment.

In addition to the above experimental lines of evidence, we have also mentioned the following lines in the results section on pages 14-15: 'Despite being regulated by Agrin, these observations suggests that MMP1 or 10 are not major downstream mediators in restoring the wound healing mechanics. Despite the fact that several MMPs are involved in cancer cell migration, matrix crosslinking, and wound healing, a concrete role of MMP12 has not been elucidated in mechanotransduction involved during cutaneous wound repair. As such, MMP12 emerged as the most significant effector for mediating Agrin's mechanotension in wound repair.' Therefore, our study revealed a new role and underlying mechanism for MMP12 to mediate the role of Agrin in wound healing.

10. Authors showed the zymography for Mmp-12 using gelatin as a product (Extended data figure 4 (b) Gelatin degradation assay detecting the catalytic activity of MMP12 in control and Agrin depleted HaCaT cell supernatants.). Generally gelatin zymography is used for the analysis of Mmp-2 and Mmp-9 activity. To analyze Mmp-12 activity casein zymography is recommended. This is interesting that using supernatant from cells culture the authors were able to show Mmp-12 gelatinolytic activity. (BIOTECHNIQUESVOL. 38, NO. 1REVIEW Zymographic techniques for the analysis of matrix metalloproteinases and their inhibitors Patricia A.M. Snoek-van Beurden & Johannes W. Von den Published Online:30 May 2018<https://doi.org/10.2144/05381RV01>)

We are grateful to this reviewer for pointing this out. Accordingly, we have tested the MMP12 activity from culture supernatants upon Agrin knockdown using casein as a substrate. The new results shown in **Fig. S9b** reveal that Agrin depletion inhibited the MMP12 catalytic activity by this recommended assay. Supplementing sAgrin to the Agrin depleted cells rescued MMP12's casein degrading activity in a dose-dependent fashion, consistent with MMP12 activity shown by gelatin degradation assay.

11. The authors should also perform the experiments on Mmp-12 KO mice and discuss how skin wound healing/re-epithelialization is accomplished in this mouse model. The use of Mmp-12 KO mice is mandatory to state that Agrin via Mmp-12 reveals its mechanotransduction.

Many thanks to this reviewer for this suggestion. Indeed, we agree that performing experiments on MMP12 knockout (KO) mouse models will enhance the value of our study. However, accessibility to MMP12 knock-out mouse models is challenging because we do not

have the mouse lines readily available at the BRC facility at A*STAR. More importantly, the MMP12 KO strain B6.129X-Mmp12tm1Sds/J (Jackson Laboratories) is also not presently available due to replenishing of cryostocks (<https://www.jax.org/strain/004855>). As such, inaccessibility to the MMP12 KO mouse strain in the near future would compromise the timely execution of this revised manuscript. Hence, we greatly appreciate a kind understanding from the reviewer in this context by accepting our alternate *in vivo* experimental model (see below).

As an alternate approach, we have used siRNA mediated suppression of dermal MMP12 *in vivo*. In this model, we depleted MMP12 by stealth siRNAs that efficiently suppressed cutaneous MMP12 levels at days 0 and 10 post-wounding (**Figs. S10a-b**). Suppression of MMP12 expression in the mouse skin delayed wound healing rates when compared to those treated with a control siRNA (**Fig. 6a-b**). More importantly, the accelerated healing rates and the degree of re-epithelization observed in sAgrin treated control animals were dramatically reduced upon MMP12 depletion within 10 days post-injury (**Fig. 6a-b**). We further noted that increased pMLC levels in the wound edges and beds of control and sAgrin treated ones were significantly attenuated in MMP12 depleted skin sections (**Fig. 6c**). In addition, sAgrin induced mature and mixed collagen fiber deposition was significantly diminished upon MMP12 depletion at the wound beds (**Fig. 6d**). Thus, MMP12 depletion renders a mechanically incompetent environment that is deprived of ECM replenishment and acto-myosin mechanotension which is no longer responsive to sAgrin. Reduced angiogenesis (fewer blood vessels) was observed within the wound beds of MMP12 depleted mouse skin (**Fig. 6e**). As such, the treatment of sAgrin failed to induce robust CD-31 expressing blood vessels when MMP12 was suppressed in the mouse skin (**Fig. 6e**). Together, these observations reveal that sAgrin requires MMP12 to generate a mechanically competent pro-angiogenic wound healing niche *in vivo*.

We sincerely hope that our thorough *in vivo* experiments would convince the reviewer that Agrin engages MMP12 to create a mechanically competent wound healing niche in the skin.

12. MTT is not a proliferation measurement assay but cell metabolic activity assay. To measure the proliferation rate ie. BrdU incorporation assay should be performed.

Thank you for pointing this out. We have performed new experiments based on BrdU incorporation and Ki67 staining for actively proliferative cells. For *in vitro* experiments, we found that a dose-dependent treatment of sAgrin enhanced the BrdU incorporation rates in mouse primary keratinocytes, suggesting that sAgrin supports proliferation and migration of keratinocytes (**Fig. S11b**). More importantly, this was also tested *in vivo* whereby the improved healing rate was supported by an enhancement of Ki67+ve proliferating cells at the base of the migrating epidermis and wound edges of sAgrin treated mouse skin sections (**Fig. S11h**). Together, these additional evidence advocate that sAgrin supports the proliferative stages of migrating keratinocytes, thereby improving the healing rates.

Likewise, we also tested the effect of Agrin depletion on the proliferative rates of keratinocytes *in vitro* via BrdU incorporation assay and *in vivo* by staining with Ki67 antibody (**Fig. S4a-c**).

References:

1. Kawahara R, Granato DC, Carnielli CM, Cervigne NK, Oliveria CE, Rivera C, *et al.* Agrin and perlecan mediate tumorigenic processes in oral squamous cell carcinoma. *PLoS One* 2014, **9**(12): e115004.
2. Chakraborty S, Lakshmanan M, Swa HL, Chen J, Zhang X, Ong YS, *et al.* An oncogenic role of Agrin in regulating focal adhesion integrity in hepatocellular carcinoma. *Nat Commun* 2015, **6**: 6184.
3. Galiano RD, Michaels Jt, Dobryansky M, Levine JP, Gurtner GC. Quantitative and reproducible murine model of excisional wound healing. *Wound Repair Regen* 2004, **12**(4): 485-492.
4. Iozzo RV, Schaefer L. Proteoglycan form and function: A comprehensive nomenclature of proteoglycans. *Matrix Biol* 2015, **42**: 11-55.
5. Gutierrez E, Groisman A. Measurements of elastic moduli of silicone gel substrates with a microfluidic device. *PLoS One* 2011, **6**(9): e25534.
6. Lin Q, Fang D, Fang J, Ren X, Yang X, Wen F, *et al.* Impaired wound healing with defective expression of chemokines and recruitment of myeloid cells in TLR3-deficient mice. *J Immunol* 2011, **186**(6): 3710-3717.
7. Sher I, Zisman-Rozen S, Eliahu L, Whitelock JM, Maas-Szabowski N, Yamada Y, *et al.* Targeting perlecan in human keratinocytes reveals novel roles for perlecan in epidermal formation. *J Biol Chem* 2006, **281**(8): 5178-5187.
8. Perrot G, Colin-Pierre C, Ramont L, Prout I, Garbar C, Bardey V, *et al.* Decreased expression of GPC1 in human skin keratinocytes and epidermis during ageing. *Exp Gerontol* 2019, **126**: 110693.

REVIEWER COMMENTS

Reviewer #1 (Remarks to the Author):

In this revised manuscript, the authors significantly improved the physiological relevance of their findings by repeating all experiments in a 3D co-culture (keratinocytes and fibroblasts) model of wounding as well as using a splinted model of mouse wound healing. Since the authors decided not to track any changes in their revised manuscript, this reviewer was unable to easily determine what was newly added to the manuscript and what was not. However, it is clear to say that most of figures 3, 6, and 7 are new, as they involve the 3D co-culture and splinted models. In this revised manuscript, the authors comprehensively show the importance of Agrin in promoting keratinocyte-specific re-epithelialization and the ability of a novel sAgrin lotion material to accelerate wound closure. They also define the Agrin-MMP12 axis in the context of wound healing. I thank the authors for putting in this tremendous work to substantially improve their manuscript.

However, I do have several suggestions and modifications that the authors should address before publication.

1) Agrin seems to play a clear role in promoting keratinocyte re-epithelialization. However, most histology images in this manuscript show clear formation of scar over the wound, shown especially by increased collagen deposition, and no signs of “tissue regeneration,” which would involve less collagen, improved biomechanical properties more similar to normal skin, and collagen architecture with similar organization to unwounded skin. In the discussion, the authors should clearly differentiate that this therapy could be used to promote tissue closure and repair, but not regeneration.

2) On line 567 in the discussion, the authors propose that this therapy could be used on “wounds reluctant to heal, such as diabetic wounds or burn injuries.” Those two types of wounds are extremely different. Diabetic wounds might indeed find benefit from sAgrin treatment, since their chronic wounds never close. Burn wounds, on the other hand, consistently result in “over-healing” in which excessive collagen deposition and tissue repair occurs, which leads to a massive scar. Since sAgrin also promotes increased collagen deposition (shown in picrosirius red analysis throughout the manuscript), sAgrin could potentially make burn patients scar even worse. This point about using sAgrin for diabetic patients but not conditions of scarring should be clearly articulated throughout the manuscript.

3) I would recommend against calling the 2D scratch assay a wound or an injury. It is misleading and inaccurate. Figure 3 is very interesting and a great figure. The new 3D co-culture “wound” model is much more physiologically relevant and could actually be called a “in vitro wound model.”

4) The authors mixed siRNA with ointment, which has interesting and translational therapeutic potential. Could the authors provide some more background on this and its potential as a translational delivery mechanism?

5) The authors splinted their wounds by gluing, but not suturing, a nylon splint around the wound. Has this method been validated before? The glue usually wears off after about one day and the wounds are released from the splinting effect. Was it reapplied daily?

Minor points:

1) Fig 1a caption says that this is “single-cell transcriptomics of wounded cells,” which seems to imply single cell RNA-sequencing, or at the very minimum 96-well plate sequencing. It seems confusing why the authors would reduce all of that dense information into a single heatmap with 5 hand picked genes. Is this bulk RNA sequencing and not single cell?

2) Figs 1b-1d and so on describe an in vitro scratch assay on a 2D plastic. There are no schematics to demonstrate the difference between 1a and 1b-d, which is confusing. Then in figure 1e, it seems like we are moving back to the in vivo mouse model. There are no clear labels on the figures and it is very laborious to have to solve the different subpanels from different experiments.

3) Figure 7e and f are nice, but the quantifications are confusing. Is the D2 graph for the non-splinted and D7 graph for splinted? Was D2 treated with Agrin or sAgrin? The x axis says the groups are “control vs Agrin.” Shouldn’t it be “BSA vs sAgrin”? There needs to be more symmetry between the D2 and D7 labels or else it is confusing.

4) Line 298 says “Fig 4e-” the dash should be removed or is it referencing multiple subpanels?

Reviewer #2 (Remarks to the Author):

I was satisfied with the revised manuscript.

Reviewer #3 (Remarks to the Author):

Authors responded to the comments. Performed by authors additional experiments and written clarification corrected and improved the manuscript. The presented work/data is convincing in the present/corrected form.

Response to reviewers' comments

We are glad to see that all three reviewers find our revised manuscript suitable for publication in Nature Communications. We are sincerely thankful for their critical and constructive suggestions that enabled us to significantly improve the study.

Reviewer #1 (Remarks to the Author):

In this revised manuscript, the authors significantly improved the physiological relevance of their findings by repeating all experiments in a 3D co-culture (keratinocytes and fibroblasts) model of wounding as well as using a splinted model of mouse wound healing. Since the authors decided not to track any changes in their revised manuscript, this reviewer was unable to easily determine what was newly added to the manuscript and what was not. However, it is clear to say that most of figures 3, 6, and 7 are new, as they involve the 3D co-culture and splinted models. In this revised manuscript, the authors comprehensively show the importance of Agrin in promoting keratinocyte-specific re-epithelialization and the ability of a novel sAgrin lotion material to accelerate wound closure. They also define the Agrin-MMP12 axis in the context of wound healing. I thank the authors for putting in this tremendous work to substantially improve their manuscript.

However, I do have several suggestions and modifications that the authors should address before publication.

Response: We appreciate the expert comments of this reviewer in helping us to improve our study. We are also delighted to see this reviewer finds our new experiments of high significance and the manuscript is overall well-suited for publication. Accordingly, we have addressed all the remaining concerns with modifications as suggested.

1) Agrin seems to play a clear role in promoting keratinocyte re-epithelialization. However, most histology images in this manuscript show clear formation of scar over the wound, shown especially by increased collagen deposition, and no signs of "tissue regeneration," which would involve less collagen, improved biomechanical properties more similar to normal skin, and collagen architecture with similar organization to unwounded skin. In the discussion, the authors should clearly differentiate that this therapy could be used to promote tissue closure and repair, but not regeneration.

Response: Thank you for this insightful suggestion. We have accordingly removed any claims in the manuscript that proposed Agrin may aid in tissue regeneration. In view of our data that Agrin treatment remodelled collagen and sustained the biomechanical properties of the healing skin, we have clearly articulated this point in our discussion section without stating any impact on potential tissue regeneration.

'In circumspection of the fact that sAgrin restores the collagen (and possibly other ECM protein) replenishment while preserving the mechano-architecture of the wound environment, we anticipate a prominent wound closure and tissue repair role associated with Agrin rather than in de novo skin tissue regeneration.'- discussion page 22.

2) On line 567 in the discussion, the authors propose that this therapy could be used on “wounds reluctant to heal, such as diabetic wounds or burn injuries.” Those two types of wounds are extremely different. Diabetic wounds might indeed find benefit from sAgrin treatment, since their chronic wounds never close. Burn wounds, on the other hand, consistently result in “over-healing” in which excessive collagen deposition and tissue repair occurs, which leads to a massive scar. Since sAgrin also promotes increased collagen deposition (shown in picrosirius red analysis throughout the manuscript), sAgrin could potentially make burn patients scar even worse. This point about using sAgrin for diabetic patients but not conditions of scarring should be clearly articulated throughout the manuscript.

Response: Once again, we appreciate this excellent scientific insight from this reviewer. As such, we hypothesize that sAgrin therapy may help to heal diabetic wounds at a faster rate by preserving the mechano-architecture and the ECM deposition rates of the healing skin microenvironment. As the reviewer suggested that burn wounds may present scarring symptoms from excess collagen and other ECM deposition, we have clearly omitted the possibility of sAgrin as therapy for burn injury patients. The discussion section has been updated as follows on page 22:

‘Hence, appropriate utilization of sAgrin as a bio-additive wound healing material may offer potential clinical value for wounds that are naturally reluctant to heal, as presented by diabetic patients. Furthermore, our study would pave the way to investigate whether Agrin levels are decreased in the skin tissues of diabetic patients and those infected with bacterial infections, presenting a basis for chronic delayed healing of wounds in these patients. Since sAgrin as bio-additive may provide considerable degree of mechanical stimuli enabling collagen remodeling and re-epithelization, we suggest that sAgrin therapy may have a potential for diabetic wound closure. Future studies involving sAgrin as a wound healing bio-additive in diabetic mouse wound healing models will also reveal the potential in future translational research among diabetic patients.’

3) I would recommend against calling the 2D scratch assay a wound or an injury. It is misleading and inaccurate. Figure 3 is very interesting and a great figure. The new 3D co-culture “wound” model is much more physiologically relevant and could actually be called a “in vitro wound model.”

Response: Thank you for this comment and for appreciating our novel 3D co-culture wound healing model. We have removed instances referring in-vitro scratch assay to be reflective of wound injury on page 4 (results section). In addition, the revised figures 1b-c has been replaced with wording ‘scratch’ instead of wound injury.

Furthermore, we have rewritten the results section as follows: ‘Because 2D-scratch assays using single cell types do not reflect the physiology of the 3D wound environment experienced during injury, we next devised an in vitro 3D-substrate stiffness dependent migration assay to recapitulate the role of Agrin in supporting keratinocytes’ migration under the simultaneous influence of underlying dermal cultures and bulk substrate rigidity as experienced by native skin tissues’ on page 9.

4) The authors mixed siRNA with ointment, which has interesting and translational therapeutic potential. Could the authors provide some more background on this and its potential as a translational delivery mechanism?

Response: Many thanks for appreciating our siRNA based approach to inhibit Agrin in the mouse skin. We fully agree with the reviewer that siRNA based topical applications may have significant translational benefits. We have introduced the following excerpt in our discussion on page 24 to illustrate the potential relevance of siRNA based therapy to study wound healing mechanisms:

‘Several studies have revealed that resuspension of siRNAs with topical formulations (such as Pluronic F127 or Aquaphore gels) may effectively inhibit target gene expression in the skin layers^{30, 66}. For instance, siRNAs against YAP and TAZ in the mouse skin have been associated with a delayed wound healing response⁶⁶. Likewise, our study employed several stealth siRNAs which were very effective in knocking down Agrin and MMP12 in the skin layers (Figures 2 and 6). Repeated topical siRNA dosage effectively inhibited the Agrin-MMP12 protein expression up to day 10 post-wounding and impacted the course of healing (Figs. S2c and S10b). We envision that conjugation of these Agrin-MMP12 siRNAs with nuclei-acid nanoparticles may improve their specificity and stability to emerge as future topical delivery agents to study skin wound healing disorders. Further research involving the conjugation of Agrin or MMP12 siRNAs with nucleic-acid based nanoparticles may be commissioned to broaden the translational impact of targeting key proteins in skin wound healing disorders.’

5) The authors splinted their wounds by gluing, but not suturing, a nylon splint around the wound. Has this method been validated before? The glue usually wears off after about one day and the wounds are released from the splinting effect. Was it reapplied daily?

Response: We appreciate the excellent insight from this reviewer. We employed a non-suture adhesive based splinting model following previously published protocols¹. Because intradermal suturing is painful to animals and time-consuming for researchers, we employed cyanoacrylate based adhesives (KrazyGlue) that have been used extensively by other research groups. Physiologically, adhesive based splinting maintained the mechanical and histological texture of wounds that was comparable to that of sutured splinted wounds¹. In addition, the adhesive based splinted wounds also showed similar wound maturation when compared to sutured splint wounds. In our studies, we further noticed that application of KrazyGlue firmly adhered the nylon splints to the mouse skin and they were not released from the wounds covered by Tegaderm. To further minimize the accidental release of splints, we applied adhesive to the lining of the splint every two days during the experimental time-frame before treatment with ointments. We have included this revised experimental set-up in our methods section under ‘punch-wound biopsy’ on page 29.

Minor points:

- a) Fig 1a caption says that this is “single-cell transcriptomics of wounded cells,” which seems to imply single cell RNA-sequencing, or at the very minimum 96-well plate sequencing. It seems confusing why the authors would reduce all of that dense information into a single heatmap with 5 handpicked genes. Is this bulk RNA sequencing and not single cell?

Response: Thank you for this comment. We apologize that this point was not well articulated in our previous version. Accordingly, we have revised our Fig 1a to simplify our analysis scheme. We analyzed publicly available single-cell RNA sequencing data ² (ref-26) that compared the gene expression profile amongst wounded versus non-wounded mouse skin keratinocytes. Among different candidate genes, we focussed on the ECM proteins/proteoglycans that were detected in this dataset and showed differential expression. The resulting gene expression of each candidate ECM protein in the wounded keratinocytes was normalized to unwounded cells on indicated day(s) post-wound injury. The differential gene expression changes of the detected ECM proteins are indicated in the heatmap (Figure 1a). We have mentioned this in our results section accordingly.

‘This work was initiated by the serendipitous discovery that the expression of several ECM proteins are altered in the lineage traced wounded mouse skin epidermal keratinocytes upon mechanical injury (26). Focusing on the early phase of the wound healing where re-epithelization is marked by Keratin 17 (KRT17) expressing keratinocytes, we detected a wound signature comprising of several ECM proteins including Agrin (AGRN), Perlecan (HSPG2), Glypican 1-3 (GPC1-3), that were enhanced in the wounded cells when compared to their non-wounded counterparts within the observed 1-10 days post-punch wound injury (Fig. 1a).’

- b) Figs 1b-1d and so on describe an in vitro scratch assay on a 2D plastic. There are no schematics to demonstrate the difference between 1a and 1b-d, which is confusing. Then in figure 1e, it seems like we are moving back to the in vivo mouse model. There are no clear labels on the figures and it is very laborious to have to solve the different subpanels from different experiments.

Response: Once again, we appreciate the reviewer’s suggestion here. As per advice, we have revised our figure panels 1a and 1e to include schematic showing the experimental workflow. Our Fig 1e schematic shows that mouse skin is analysed by histology (Fig 1e-confocal immunohistochemistry and Fig 1f: immunohistochemistry) on indicated days post-punch wound biopsy. Moreover, our revised Fig 1a also shows that we have analysed the publicly available single-cell RNA sequencing results (dataset E-MTAB-6583) comparing non-wounded versus wounded mouse keratinocytes. The respective figure legends have also been modified on page 45. These modifications clearly illustrate that Figures 1a and 1e-f focus on mouse skin tissue analysis, while Figures 1b-d used skin cells as model systems.

- c) Figure 7e and f are nice, but the quantifications are confusing. Is the D2 graph for the non-splinted and D7 graph for splinted? Was D2 treated with Agrin or sAgrin? The x axis says the groups are “control vs Agrin.” Shouldn’t it be “BSA vs sAgrin”? There needs to be more symmetry between the D2 and D7 labels or else it is confusing.

Response: We apologize for not labelling these figure panels clearly and sincerely thank the reviewer for pointing this out. Accordingly, we have revised our Figure 7. The updated figure 7e-f shows that quantified MMP12 and pMLC are labelled as non-splinted d2 and splinted d7, respectively. Moreover, we have also redone the quantifications comparing BSA (as control) and sAgrin for all the figure panels to maintain consistency across all the panels in the figure.

d) Line 298 says “Fig 4e-” the dash should be removed or is it referencing multiple subpanels?

Response: Many thanks for pointing this out. We have removed this typological error.

Reviewer #2 (Remarks to the Author):

I was satisfied with the revised manuscript.

Response: We sincerely appreciate the reviewer's critical insights that have improved the significance of our study. Thank you.

Reviewer #3 (Remarks to the Author):

Authors responded to the comments. Performed by authors additional experiments and written clarification corrected and improved the manuscript. The presented work/data is convincing in the present/corrected form.

Response: We are delighted to find that this reviewer finds our revised manuscript acceptable for publication. Many thanks for the insightful suggestions and supporting our study.

References:

1. Sabol F, Vasilenko T, Novotny M, Tomori Z, Bobrov N, Zivcak J, *et al.* Intradermal running suture versus 3M Vetbond tissue adhesive for wound closure in rodents: a biomechanical and histological study. *Eur Surg Res* 2010, **45**(3-4): 321-326.
2. Joost S, Jacob T, Sun X, Annusver K, La Manno G, Sur I, *et al.* Single-Cell Transcriptomics of Traced Epidermal and Hair Follicle Stem Cells Reveals Rapid Adaptations during Wound Healing. *Cell Rep* 2018, **25**(3): 585-597 e587.

REVIEWERS' COMMENTS

Reviewer #1 (Remarks to the Author):

I appreciate that the authors have answered all of my questions and revised the manuscript to reflect those answers. I have no further concerns.

Reviewer #1 (Remarks to the Author)

I appreciate that the authors have answered all of my questions and revised the manuscript to reflect those answers. I have no further concerns.

Response: We are grateful for the constructive suggestions provided by this reviewer that has greatly strengthened our study. We also thank all our reviewers for their time and efforts in reviewing our manuscript.